# Multiple Choice Learning of Low Rank Adapters for Language Modeling

## Abstract

We propose `LoRA-MCL`, a training scheme that extends next-token prediction in language models with a method designed to decode diverse, plausible sentence continuations at inference time. Traditional language modeling is an intrinsically ill-posed problem: given a context, multiple "futures" may be equally plausible. Our approach leverages Multiple Choice Learning (MCL) and the Winner-Takes-All loss to efficiently handle ambiguity through Low-Rank Adaptation. We provide a theoretical interpretation of applying MCL to language modeling, assuming the data is generated from a mixture of distributions. To illustrate the proposed approach, we use data sampled from mixtures of Markov chains. We then demonstrate with experiments on visual and audio captioning, as well as machine translation, that our method achieves high diversity and relevance in generated outputs.

## 1 Introduction

Predicting what a person will say next or describing the content of an audio or visual scene with text is difficult, if not impossible, to do with perfect accuracy. When the context is not informative enough, external factors may lead to different scenarios or *modes* of plausible text continuations [96, 10, 56]. In such ambiguous tasks, the conditional distribution over the space of output sentences given the input context may be multi-modal due to the underlying inherent uncertainty [51].

Initially proposed for text processing, transformer-based auto-regressive language models [67, 68] have quickly become a general framework for modeling streams of tokens, which can also represent, for instance, images and audio signals [81, 6, 43]. Such models are trained as next-token predictors and allow, by nature, for addressing this uncertainty, by generating plausible output sentences through a wide body of maximum a posteriori (MAP) or sampling-based decoding approaches [87]. While sampling allows exploration and diversity, it may lead to unreliable responses and requires truncation to avoid unexpected answers, partly due to the overestimation of low-probability tokens [98, 24]. When seeking reliable and expected answers, MAP estimation techniques, like Beam Search [46], look for sentences that maximize the model's likelihood. However, these alternatives have drawbacks as they may lack diversity, be prone to repetition loops [33], and may sound unnatural [25]. Some approaches, e.g., Diverse Beam Search (DBS) [84], were therefore proposed to artificially increase the diversity at inference, e.g., through a diversity penalty parameter $\lambda$, to find a tradeoff between generation quality and sample diversity [79]. In contrast with these methods, our approach aims to *predict* diverse sentences reflecting the ambiguity of the input context.

Multiple Choice Learning (MCL) [20, 38] has emerged as a paradigm for addressing ambiguous tasks. It generally consists of a network with a shared backbone and multiple output heads. During training, it utilizes the winner-takes-all loss for adaptively updating the head that performs the best for each example. This is a competitive training scheme that specializes each model to subsets of the conditional output distribution [73]. In this paper, we propose incorporating this idea for language model finetuning, leveraging multiple Low-Rank Adapters [27] instead of multiple heads, which may be impractical due to computation requirements and architectural constraints. Our method natively generates diverse and plausible sequences in a single forward pass, aiming to best approximate the conditional output distribution.

Our contributions are as follows:

**We propose a new paradigm that adapts MCL for token sequence modeling**, with a method, `LoRA-MCL`, that is particularly suited for efficient finetuning of language models.

**We provide a theoretical analysis of our approach, which applies MCL to language modeling.** Assuming the sequences are sampled from a mixture of distributions, we explain in particular why `LoRA-MCL` should enable capturing the modes of the data distribution. We then consider the case of Markov chains, for which we validate our claims with a well-designed toy example.

**We conduct an extensive experimental study validating our approach** on audio and vision captioning tasks, as well as machine translation, demonstrating wide applicability in challenging real-world scenarios and showing an excellent diversity–quality trade-off.[1]

## 2 PROBLEM SETUP

Let $x \triangleq (x_t)_{t=1}^T \in \mathcal{V}^T$ be a sequence of $T$ tokens belonging to a finite vocabulary $\mathcal{V} = \{1, \ldots, |\mathcal{V}|\}$, and $c \triangleq (c_t)_{t=1}^\tau \in (\mathbb{R}^d)^\tau$ be a sequence of $\tau$ context vector embeddings of dimension $d$. Language modeling aims at learning the law $p(x \,|\, c) = \prod_{t=1}^T p(x_t \,|\, x_{<t}, c)$ using a model $p_\theta$ with parameters $\theta$, by minimizing the following negative log-likelihood loss, which is equivalent to the maximum likelihood estimation (MLE):

$$\mathcal{L}(\theta) = -\mathbb{E}_{c,x}[\log p_\theta(x \,|\, c)] = \mathbb{E}_{c,x}\Big[ -\sum_{t=1}^T \log p_\theta(x_t \,|\, x_{<t}, c)\Big], \tag{1}$$

where $x_{<t}$ denotes the sequence of tokens prior to time $t$.

Optimizing (1) is referred to as *teacher-forcing* [89], where $p_\theta$ is fed with target (instead of predicted) tokens $x_{<t}$ during training [80]. When using a transformer architecture [82], this is implemented with causal attention modules, which allow for computing the conditional distributions in parallel through all time steps within a single forward pass.

During inference, decoding methods [87] proceed to generating sequences $\hat{x}$ from the trained model $p_\theta$ in an auto-regressive fashion. First, they start with a conditional distribution for the first token from the context: $p_\theta(x_1 \,|\, c)$, which allows selecting $\hat{x}_1$. Then, for $t \geq 2$ they predict $p_\theta(x_t \,|\, \hat{x}_{<t}, c)$, and select $\hat{x}_t$, until reaching either the sequence length limit or an end-of-sentence (EOS) token. The choice of the decoding method used to generate $K$ candidate sequences $\hat{x}^1, \ldots, \hat{x}^K$ depends on the purpose of the task, but the general goal is $(i)$ to get highly likely sentences, i.e., ones that maximize $p_\theta(\hat{x} \,|\, c)$; and $(ii)$ to get sufficiently diverse sentences, as can be measured by $n$-gram similarity [28]. Although this is a widely adopted paradigm, we show next how this training and decoding pipeline can be improved: instead of *artificially* generating diversity at inference time, we aim at *learning to predict* sequences $\hat{x}^1, \ldots, \hat{x}^K$ that cover well the modes of the target distribution $p(x \,|\, c)$.

## 3 METHODOLOGY

### 3.1 MOTIVATION

In language modeling, topic models [62, 5, 10] are data-generating processes in which the ground-truth probability distribution of sequences is modeled as a mixture of latent components or topics. For example, the sequence "I am eating ..." may have multiple plausible continuations, but the likelihood of each depends heavily on contextual factors such as the speaker's location, which influences their culinary habits. Each location (or context) can thus be associated with a distinct word distribution. In topic models, data generation proceeds by first sampling a topic $z \in \mathcal{Z}$ for each sentence (usually referred to as a *document* in the literature), and then sampling words (or $n$-grams) from the distribution associated with that topic.

With this in mind, MLE in (1) may not be suitable [96]. While MLE is effective for estimating the overall distribution $p(x)$, it does not capture the individual components when it is expressed as a mixture, i.e., $p(x) = \sum_k p(z_k)p(x \,|\, z_k)$. In such cases, MLE tends to model the aggregate rather than distinguish topic-specific distributions $p(x \,|\, z_k)$.

---

[1]The code will be made publicly available.

### 3.2 APPLYING MULTIPLE CHOICE LEARNING TO LANGUAGE MODELING

Our approach is inspired by the multiple choice learning (MCL) literature [20, 38]. We propose the following training scheme, intending to enable the recovery of the different topics $z_k$. Instead of a single model, we consider a *set* of models $(\theta_1, \ldots, \theta_K)$. Then the objective (1) is replaced by one consisting of iterating between the following two steps:

1. For each training sample $(c, x)$ in the batch $\mathcal{B}$: Compute $p(x \mid c; \theta_k)$ for $k \in \{1, \ldots, K\}$, and choose the best model $k^\star(x, c) = \mathrm{argmax}_k \, p(x \mid c; \theta_k)$.

2. Compute the winner-takes-all (WTA) loss as:

$$\mathcal{L}^{\mathrm{WTA}}(\theta_1, \ldots, \theta_K) = -\mathbb{E}_{c,x}\left[\max_{k=1,\ldots,K} \log p(x \mid c; \theta_k)\right], \qquad (2)$$

where $\log p(x \mid c; \theta_k) = \sum_{t=1}^{T} \log p(x_t \mid x_{<t}, c; \theta_k)$, and perform an optimization step.

This training procedure, similar to a hard-EM style optimization [58, 88], is a competitive training scheme that encourages the different models to explore different areas of the data distribution. However, it is subject to two main issues: First, using $K$ models instead of a single one drastically increases the training time and memory cost, which may be intractable for large language models (LLMs). Second, the optimization may be subject to collapse, where the same models are chosen as winners through the iterations, leaving the other models untrained. In the next section, we describe how we solve these issues with our approach `LoRA-MCL`.

### 3.3 `LoRA-MCL` METHOD

Multiple choice learning typically alleviates the high training cost issue of $K$ models by training a single model with several heads [38, 37]. However, we argue that such an approach is not well-suited for fine-tuning language models. First, heads of most language models are quite large (for example, in Qwen2-Audio [6] the `lm_head` has $d \times |\mathcal{V}| = 4096 \times 156032 \simeq 640\mathrm{M}$ parameters), and standard MCL would not scale easily with the number of heads. Second, the initialization of the heads poses several challenges. Initializing each head with the parameters of the head of the pretrained model requires special care, as the collapse of the predictions is likely given the very similar hypotheses (same parameters). A complete re-initialization of the heads is detrimental to performance, as numerous training iterations would be necessary to reach the same level of knowledge as in the pretrained head. For these reasons, we consider a Low Rank Adapter (LoRA) approach [27] due to its excellent trade-off between performance and computational requirements, as well as its wide adoption in the context of large model fine-tuning.

Let $\theta$ be the parameters of the pretrained base model. At each layer $\ell$ where LoRA is enabled, we use a family of adapters $(A_\ell^k, B_\ell^k) \in \mathbb{R}^{d \times r} \times \mathbb{R}^{r \times d}$ for $k \in \{1, \ldots, K\}$. Let

$$\theta_k = \theta \cup \left\{(A_\ell^k, B_\ell^k) \mid \ell = 1, \ldots, L\right\}, \qquad (3)$$

be the set of parameters that are involved in hypothesis $k$, with $L$ being the total number of layers where LoRA is used. Training in the WTA fashion involves computing $p(x \mid c; \theta_k)$ for $k = 1, \ldots, K$. To avoid situations where some heads may be under-trained, including the *collapse* when a single head is trained, we use the relaxation of the winner-takes-all training objective of the form

$$\mathcal{L}^{\mathrm{WTA}}(\theta) = -\mathbb{E}_{c,x}\left[\sum_{k=1}^{K} q_k \log p(x \mid c; \theta_k)\right]. \qquad (4)$$

where $\{q_k\}$ is a set of positive coefficients that sum to 1. These coefficients assign higher weight to the winning head $q_{k^\star}$ while still providing nonzero gradient contributions to the other heads, thereby mitigating collapse. We experimented with two relaxation techniques. First, *Relaxed-WTA* [73] where $q_{k^\star} = 1 - \varepsilon$ and $q_k = \frac{\varepsilon}{K-1}$ for $k \neq k^\star$, with $\varepsilon > 0$ a small constant. We also considered the *annealed MCL* method [66], which introduces a temperature parameter $\tau$:

$$q_k(x, c; \tau) = \frac{p(x \mid c; \theta_k)^{\frac{1}{\tau}}}{Z_{x,c}(\tau)}, \qquad Z_{x,c}(\tau) = \sum_{s=1}^{K} p(x \mid c; \theta_s)^{\frac{1}{\tau}}. \qquad (5)$$

Here the temperature $t \mapsto \tau(t)$ follows a decreasing schedule, typically $\tau(t) = \tau(0)\rho^t$ with $\rho < 1$ and $\tau(0) > 0$. At high temperatures, training is distributed more evenly across all hypotheses, preventing collapse; as $\tau \to 0$, the method converges to the greedy WTA regime.

### 3.4 ACCELERATING LoRA-MCL TRAINING WITH PARALLELIZATION OVER THE HYPOTHESES

A naive implementation of LoRA-MCL would require looping over the $K$ hypotheses in the batch to evaluate each candidate separately, which would drastically slow down training. To avoid this, we process all hypotheses in parallel. Specifically, given an input sequence, we duplicate it $K$ times along the batch dimension. Each copy is then passed through a LoRA-adapted transformer, but crucially, we implement this in a *grouped fashion*: instead of running $K$ independent forward passes, we combine them into a single batched operation where each group corresponds to one hypothesis. In practice, this can be achieved using a grouped 1D convolution (nn.Conv1d in PyTorch) with $K$ groups, so that each hypothesis uses its own LoRA weights while still sharing the frozen base model. This trick effectively multiplies the batch size by $K$ while keeping the memory overhead manageable (since the LoRA rank $r \ll d$). It removes the sequential loop, enabling efficient parallel training of all hypotheses. Further implementation details are provided in Appendix E.5.

## 4 THEORETICAL ANALYSIS

We justify the use of MCL for language modeling by assuming the sequence distribution is a mixture. We contrast this formulation with standard MLE (1). Section 4.1 links our method to EM and derives lower and upper bounds on the optimal achievable test loss. We then apply this analysis to the case of Markov chains and simulate the method's dynamics in a controlled setting.

### 4.1 TRAINING DYNAMICS AND OPTIMALITY CONDITIONS

For the next-token prediction loss in (1), one can show that: $\min_\theta \mathcal{L}(\theta) = \mathcal{H}(x \mid c)$, where $\mathcal{H}(x \mid c) \triangleq -\mathbb{E}_{c,x}[\log p(x \mid c)]$ is the entropy of $p$ [47]. Following the rationale of Section 3.1, let us now assume the data distribution can be written as a mixture. In the case of the WTA loss under this setup, we have the following proposition.

**Proposition 1** (Proof in Appendix B). *Assume the data-generating process is $p(x \mid c) = \sum_{k=1}^{K} p(z_k \mid c)\, p(x \mid z_k, c)$ (Asm. 2). Also assume perfect model expressiveness (Asm. 1), and large enough batch size to approximate the true risk well (Asm. 3). Then:*

*(i) Winner-Takes-All training in LoRA-MCL acts as a conditional form of the hard-EM.*

*(ii) Assuming disjoint components (Asm. 4), and assuming (with one permutation) that $p(x \mid z_k, c) = p(x \mid c; \theta_k)$ for each $k$, $\mathcal{L}^{\mathrm{WTA}}(\theta) = -\mathbb{E}_{x,c}\left[\max_{k=1,\dots,K} \log p(x \mid c, z_k)\right]$, and:*

$$\mathcal{L}^{\mathrm{WTA}}(\theta) = \mathcal{H}(x \mid c, z) \triangleq \mathbb{E}_c\left[\sum_{k=1}^{K} p(z_k \mid c)\mathcal{H}\big(x \mid c, z_k\big)\right], \qquad (6)$$

*where $\mathcal{H}(x \mid c, z)$ is the conditional entropy given the random variable $z$.*

*(iii) We have the following inequalities:*

$$\min_\theta \mathcal{L}(\theta) - \log K \overset{(a)}{\leq} \min_\theta \mathcal{L}^{\mathrm{WTA}}(\theta) \overset{(b)}{\leq} \mathcal{H}(x \mid c, z) \overset{(c)}{\leq} \min_\theta \mathcal{L}(\theta), \qquad (7)$$

*where $\min_\theta \mathcal{L}(\theta) = \mathcal{H}(x \mid c)$.*

$(i)$ in Proposition 1 describes the relationship between LoRA-MCL and the hard-EM algorithm. $(ii)$ provides an expression for the WTA loss as a conditional entropy, $\mathcal{H}(x \mid c, z)$, under the assumption of a perfect matching between the hypotheses and the modes. Proposition 1 also establishes in $(iii)$ both a lower and an upper bound on the optimal achievable loss for LoRA-MCL, given in $(a)$ and $(b)$, respectively. Note that the gap between these bounds is $\mathcal{H}(x \mid c, z) - \min_\theta \mathcal{L}(\theta) + \log K = -\mathbb{E}_c[\mathcal{I}(x, z \mid c)] + \log K$ where $\mathcal{I}$ denotes the mutual information. Finally, $(c)$ shows in particular that $\min_\theta \mathcal{L}^{\mathrm{WTA}}(\theta) \leq \min_\theta \mathcal{L}(\theta)$.

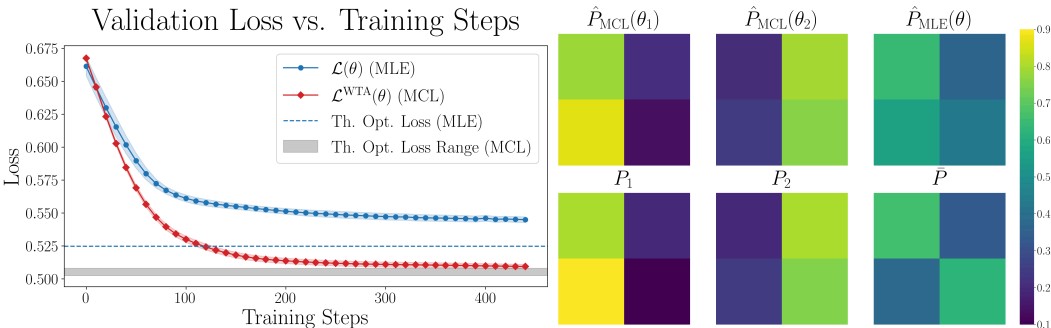

Figure 1: **Comparison of MCL with MLE.** *(Left)* Validation loss over training steps (averaged across three random seeds) for `LoRA-MLE` (blue) and `LoRA-MCL` (red). The theoretical optimal MLE loss, approximated via Monte Carlo sampling, is given by the entropy $\mathcal{H}(x)$. The gray shaded region represents the lower and upper bounds of the theoretical optimal MCL loss, as given by $(a)$ and $(b)$ in (7). *(Right)* Learned transition matrices (top) are compared with the references (bottom). MLE converges approximately toward the weighted average $\bar{P}$ defined as the right-hand side of (9). In contrast, `LoRA-MCL` successfully recovers the two underlying modes.

### 4.2 CASE OF MARKOV CHAINS

To make the analysis more concrete, we consider the case where the sequence of tokens is generated from Markov chains. Formally, given a finite state space $\mathcal{V}$, a homogeneous Markov chain is defined by an initial distribution $\pi$ over $\mathcal{V}$ and a transition matrix $P \in [0,1]^{\mathcal{V} \times \mathcal{V}}$. A sequence $(x_1, \ldots, x_T)$ is sampled from the Markov chain if $x_1 \sim \pi$ and, for each $t \geq 1$, the conditional distribution of $x_{t+1}$ given $x_t$ is $x_{t+1} \mid x_t \sim P_{x_t, \cdot}$, where $P_{i,j} = \mathbb{P}(x_{t+1} = j \mid x_t = i)$. In the following, we ignore the initial warm-up phase and we assume that $\pi$ is the stationary distribution of $P$, i.e., such that $\pi P = \pi$. In this case, we will denote $x \sim \mathrm{MC}(P)$.

While the study of the training dynamics of transformers on Markov Chain data has been investigated in previous works [13, 70, 50, 97], our setup instead considers a **mixture** of Markov chains [19, 32]. Assuming a uniform mixture, the data generating process is $x \sim \frac{1}{K} \sum_{k=1}^{K} \mathrm{MC}(P_k)$;

$$z_k \sim \mathcal{U}(1, \ldots, K), \quad x \mid z_k \sim \mathrm{MC}(P_k) . \tag{8}$$

When training a language model on such sequences, we have the following Corollary from Prop. 1, where the context $c$ is ignored for simplicity.

**Corollary 1** (Proof in Appendix C). *Let us assume that the data-generating process is a uniform mixture of first-order Markov chains. Let $\hat{P}(\theta) \triangleq (p(x_{t+1} = j \mid x_t = i))_{i,j}$ be the predicted transition matrix using a language model with parameters $\theta$. Under the same Asm. as in Prop. 1:*

(i) *Whenever the MLE estimator trained with (10) reaches its optimal loss, we have*

$$\hat{P}(\theta)_{i,j} = \frac{1}{\sum_{s=1}^{K} (\pi_s)_i} \sum_{k=1}^{K} (\pi_k)_i (P_k)_{i,j} , \tag{9}$$

*where $\pi_k \in [0,1]^{\mathcal{V}}$ is the stationary distribution of $P_k$.*

(ii) *The inequality in (7) holds in this context, where the conditional entropy $\mathcal{H}(x \mid z)$ can be computed by a weighted average of the entropy rate of each Markov Chain.*

The entropy $\mathcal{H}(x)$, which is the theoretical optimal loss of the MLE baseline, can be computed either exactly for short sequences or approximated, e.g., through Monte-Carlo integration. Note that we considered first-order Markov chains in our analysis. We expect the properties to generalize to higher orders. Please refer to Appendix C for further discussions.

### 4.3 ILLUSTRATION WITH SYNTHETIC DATA

To illustrate our approach, let us evaluate our algorithm on a synthetic dataset, for which results are given in Figure 1. We used $\mathcal{V} = \{1, 2\}$, and $(P_1, P_2) = (P(p_1, q_1), P(p_2, q_2))$, with $P(p, q) \triangleq$

$$\begin{bmatrix} 1-p & p \\ q & 1-q \end{bmatrix}$$ and $p, q \in [0, 1]$. We sampled data from a mixture of two Markov chains following (8): we generated a sequence by sampling first $P_k$ with $k \sim \mathcal{U}\{1, 2\}$. Once $P_k$ was set, we sampled uniformly the initial state, then sampled the Markov chain according to the transition matrix until reaching the maximum sequence length ($T = 32$ here).

We considered a GPT-2-like architecture [68, 4] using local-attention suggested by Makkuva et al. [50] to improve convergence on Markov chain data. We then trained the model with 1 and 2 hypotheses, using LoRA adapters as described above, 1 hypothesis corresponding to vanilla maximum likelihood estimation. Training details are provided in Appendix D. Figures 1 and 5 show both the evolution of the losses along training and the predicted transition matrix from the trained models. We see that the optimum is close to being global in this setup. While `LoRA-MCL` matches can capture the two modes of the mixture, we see that MLE tends to predict the weighted average of the transition matrices given by (9), which is consistent with Prop. 1.

## 5 EMPIRICAL EVALUATION

We evaluate `LoRA-MCL` on realistic datasets and large-scale models for audio and image captioning tasks, as well as machine translation. Predicting a textual description for images or audio signals is an ill-posed problem: from an input image or audio clip, multiple descriptions may be plausible: this is a real-world case where the conditional output distribution is inherently multi-modal. Similarly, Machine Translation is a one-to-many problem [63, 56]. We demonstrate that `LoRA-MCL` provides a competitive approach for capturing these distribution when fine-tuning either audio vision-language or language-only models. We present hereafter the experimental setup. See Apx. E.6 for details.

### 5.1 EXPERIMENTAL SETUP IN CAPTIONING

**Datasets.** We experimented on both Clotho-V2 [16, 12], and AudioCaps [18, 34] datasets for the audio captioning task, while we make use of the TextCaps dataset [78] for the challenging task of image captioning with reading comprehension. Table 4 describes the datasets sizes.

**Experimental details in audio.** We used the instructed Qwen2-Audio [6] as the base model, which has $\sim$8.4 billion parameters and a vocabulary size $|\mathcal{V}| = 156,032$. We used LoRA adapters applied to the $Q, K, V$ linear projections of the attention modules, and the upside and downside projections of the feedforward blocks, across all layers. We used a rank $r$ and scaling factor $\alpha$, with $r = \alpha = 8$ unless otherwise stated. We trained for 1 and 10 epoch on AudioCaps and Clotho respectively.

**Experimental details with visual data.** We used LLaVA 1.6 [43], as the base model which features $\sim$7.1 billion parameters and a vocabulary size $|\mathcal{V}| = 32,000$. We applied LoRA adapters only for the LLM decoder following [104]. The adapters were applied to $Q, K, V$, upside and downside projections as in Qwen2-Audio, and we used $r = \frac{\alpha}{4} = 8$ unless otherwise stated. Training was done over 1 epoch (without validation data), and the validation set of TextCaps was used for evaluation.

**Metrics.** We evaluate both quality and diversity. For quality, we report test-loss and standard NLG metrics (BLEU, ROUGE, METEOR) [64, 42, 3], and captioning-specific scores (CIDEr, SPICE, and SPIDEr) [83, 1, 44], which better correlate with human judgments in captioning. We also use Sentence-BERT [72]. We consider sentence-based oracle evaluation for these metrics (see [38, 36]). For diversity, we used mBLEU-4 [56] measuring similarity across generated captions [106].

**Baselines.** To validate our approach, we compare it against the MLE baseline (`LoRA-MLE`) trained using (1), under the same conditions as the ones considered for our multi-hypothesis model. Specifically, both models use the same LoRA configuration, the same number of trainable parameters (the LoRA rank for the baseline is $K\times$ larger to this end), and the same number of iterations. We also considered a Mixture of Low Rank Experts [59, 91, 41] (`LoRA-MoE`) as baseline. See Apx. E.3 for more details on the training methods. At inference time, for each decoding method applied to the baseline that returns $K$ sentences, we decode the same number of candidates with `LoRA-MCL`. When evaluating MAP methods such that greedy, beam search (BS) [46] and diverse beam search (DBS) [84], we ensure a consistent computational budget by aligning the number of forward passes. In this case, if `LoRA-MLE` or `LoRA-MoE` uses a beam size of $B$, our model uses a beam size of $\frac{B}{K}$ per hypothesis. Finally, we experimented Test-Time Augmentation (TTA) with `LoRA-MLE` in Audio Captioning, applying SpecAugment [65] $K$ times to the input Spectrogram to expect diverse outputs.

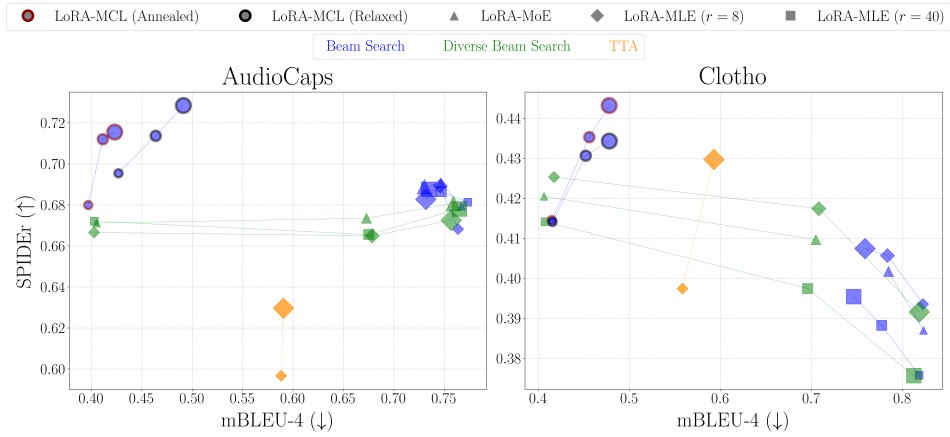

Figure 2: **Quality–diversity trade-off on audio captioning (5 candidates).** SPIDEr ($\uparrow$) for quality, mBLEU-4 ($\downarrow$) for diversity. Marker shape stands for the method, color for the decoding method, size is proportional to forward passes per example at inference. `LoRA-MLE` uses $r \in \{8, 8K\}$ for parameter parity. `LoRA-MCL` uses circle markers: Relaxed (black edge) and Annealed (red edge).

## 5.2 AUDIO CAPTIONING

**Quality vs. diversity trade-off.** Quality–diversity performance is shown in Figure 2. For readability, only the TTA runs (orange diamonds) with optimal augmentation strength on the quality–diversity are displayed front in Figure 2. It is pretty strong on Clotho, but performs poor quality on AudioCaps. We notice that a well-chosen value of $\lambda$ allows DBS applied to `LoRA-MLE` and `LoRA-MoE` to be competitive. `LoRA-MCL` (circles) achieves the best trade-off between quality and diversity, appearing in the top-left corner of the plot, where the best relaxation technique (annealed or relaxed) depends on the setup. Although increasing the beam size generally improves performance for standard beam search, we observe that increasing the beam size within each group in DBS can negatively impacts DBS, as observed in Clotho. Full results are in Tables 5 and 7.

**Effect of the number of hypotheses.** Table 1 reports the test negative log-likelihood (NLL) as a function of the number of hypotheses, with `LoRA-MCL` trained using $\varepsilon = 0.05$ and $r = 8$. The monotonically decreasing trend provides further evidence for Proposition 1, indicating that `LoRA-MCL` achieves better coverage of the data distribution modes as $K$ increases.

Table 1: **Test Loss ($\downarrow$) as a function of $K$.**

| Training | $K$ | AudioCaps | Clotho |
|---|---|---|---|
| `LoRA-MLE` ($r = 8$) | 1 | 2.203 | 2.812 |
| `LoRA-MLE` ($r = 40$) | 1 | 2.181 | 2.910 |
| `LoRA-MCL` | 2 | 2.096 | 2.692 |
| `LoRA-MCL` | 3 | 2.063 | 2.663 |
| `LoRA-MCL` | 5 | 1.999 | 2.643 |
| `LoRA-MCL` | 7 | **1.932** | **2.612** |

Table 2: **Quality and Diversity Evaluation on TextCaps (3 candidates).** Best in **bold**; second-best underlined. Higher is better ($\uparrow$) except mBLEU-4 ($\downarrow$), which measures diversity. `LoRA-MCL` is trained with $\varepsilon = 0.1$, $r = 8$ and $\alpha = 32$. `LoRA-MLE` is trained with $r = 24$, $\alpha = 96$. In the rows with $^\dagger$ we trained `LoRA-MLE` with $r = 8$ and $\alpha = 32$.

| Training | Decoding | Beam | mBLEU$_4$ | BLEU$_4$ | METEOR | sBERT | CIDEr$_D$ | SPICE | SPIDEr |
|---|---|---|---|---|---|---|---|---|---|
| `LoRA-MLE` | BS | 3 | 0.688 | 0.318 | 0.315 | 0.670 | 1.517 | 0.244 | 0.873 |
| `LoRA-MLE` | BS | 6 | 0.786 | 0.338 | 0.326 | 0.671 | 1.557 | 0.246 | 0.895 |
| `LoRA-MLE` | DBS ($\lambda = 0.8$) | 3 | 0.437 | 0.349 | 0.327 | 0.686 | 1.590 | 0.251 | 0.909 |
| `LoRA-MLE` | DBS ($\lambda = 1.0$) | 3 | **0.416** | 0.348 | 0.326 | 0.685 | 1.586 | 0.250 | 0.906 |
| `LoRA-MLE` | DBS ($\lambda = 0.8$) | 6 | 0.671 | 0.341 | 0.328 | 0.681 | 1.573 | 0.251 | 0.903 |
| `LoRA-MLE` | DBS ($\lambda = 0.8$)$^\dagger$ | 3 | 0.531 | 0.346 | 0.326 | 0.685 | 1.589 | 0.255 | 0.912 |
| `LoRA-MLE` | DBS ($\lambda = 1.0$)$^\dagger$ | 3 | 0.425 | 0.357 | 0.327 | 0.686 | 1.601 | 0.252 | 0.915 |
| `LoRA-MoE` | DBS ($\lambda = 0.8$) | 3 | 0.441 | 0.353 | 0.327 | 0.685 | 1.616 | 0.254 | 0.924 |
| `LoRA-MoE` | DBS ($\lambda = 1.0$) | 3 | 0.421 | 0.354 | 0.328 | 0.685 | 1.622 | 0.253 | 0.926 |
| `LoRA-MoE` | DBS ($\lambda = 0.8$) | 6 | 0.678 | 0.349 | 0.330 | 0.680 | 1.608 | 0.255 | 0.922 |
| `LoRA-MCL` | BS | 1 | 0.520 | 0.344 | 0.330 | **0.690** | **1.674** | 0.255 | **0.955** |
| `LoRA-MCL` | BS | 2 | 0.490 | **0.360** | **0.333** | 0.687 | 1.627 | **0.258** | 0.932 |

Table 3: **SPIDEr ($\uparrow$) & mBLEU-4 ($\downarrow$) on different parts of synthetic test set.**

| Test subset | Training | SPIDEr | mBLEU-4 |
|---|---|---|---|
| French | LoRA-MLE | 0.411 | 0.138 |
| | LoRA-MCL | **0.464** | **0.027** |
| English | LoRA-MLE | **0.756** | 0.126 |
| | LoRA-MCL | 0.722 | **0.029** |

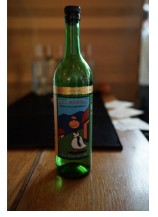

**LoRA-MLE.**
{A bottle of Cerveza is on a table.}
{Une bouteille de vin de cidre de cidre de cidre [...]}
**LoRA-MCL.**
{A bottle of beer with a label that says "Sel Maguet"}
{Une bouteille de vin est étiquetée avec le mot « Maguay ».}

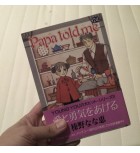

**LoRA-MLE.**
{A book titled Papa Told Me is being held by a person.}
{A book called Papa told me is being held by a person.}
**Lora-MCL.**
{A book titled Papa Told Me is being held by a person}
{Un livre papier intitulé Papa Told Me.}

Figure 3: **Observing specialization in bilingual image description.** Quantitative (*Left*) and Qualitative *(Right)* analysis for LoRA-MLE and LoRA-MCL in the setup of Section 5.3.2.

## 5.3 IMAGE DESCRIPTION WITH READING COMPREHENSION

### 5.3.1 QUALITY AND DIVERSITY EVALUATION

Evaluation in image captioning Table 2 confirms the trends observed in the audio captioning task. At an equal number of forward passes, LoRA-MCL outperforms LoRA-MLE and LoRA-MoE, even using DBS with a $\lambda$ diversity parameter specifically optimized for the task (SPIDEr of 0.955 against 0.926). Consistently with audio captioning, increasing the number of beams in each group can decrease diversity and does not improve the performance of LoRA-MLE and LoRA-MoE. However, we noticed that the DBS with LoRA-MLE tends to generate more diverse outputs than the greedy decoding of LoRA-MCL (mBLEU of 0.416 against 0.520). Combining DBS with LoRA-MCL in the future may be an option to address this.

### 5.3.2 OBSERVING HYPOTHESIS SPECIALIZATION IN BILINGUAL IMAGE DESCRIPTION

To highlight the behavior of LoRA-MCL in a realistic case where one can control the modes of the data-generating process, we simulated an artificial bi-modal distribution of the dataset, similarly to the setup of the toy experiment of Section 4.3. We did so by translating half of the captions of the data from English to French (using T5-small [69]), while keeping the prompts in English.

We trained a 2-hypothesis LoRA-MCL model and LoRA-MLE baseline. We observed a specialization of each hypothesis towards a given language (one hypothesis learned French and the other English): at test-time, the winning head is the first one in $\sim 89\%$ of the French captions and the second one in $\sim 97\%$ of the English captions. Table 3 reports quality/diversity on the synthetic test set. LoRA-MCL uses greedy decoding, LoRA-MLE DBS with $\lambda = 0.8$ (maximizing its performance). Overall performance is similar, but LoRA-MCL is notably more diverse (mBLEU-4: FR 0.027 vs 0.138, EN 0.029 vs 0.126) and outperforms on French (SPIDEr 0.464 vs 0.411), with a slight reduction in English performance. Consistently with Sec. 4.2, LoRA-MLE learns an average of the two modes (likely biased to English from LLaVA pretraining), whereas LoRA-MCL separates them.

Fig. 3 qualitatively illustrates the behavior of the models: LoRA-MLE learns a weighted average of the two modes, shifted towards English, and it is sometimes not able to output French captions within the 2 candidates as in the book example. We found LoRA-MLE to be particularly prone to errors when outputting French sentences: for instance, in the second generation of examples with an image of a bottle, LoRA-MLE enters a repetition loop. On the other hand, LoRA-MCL is less affected by those artifacts on French sentences, and successfully captures the two modes of the distribution, benefiting from the specialization of each hypothesis.

## 5.4 DIVERSE MACHINE TRANSLATION

We evaluate LoRA-MCL for zero-shot machine translation (MT) with LLMs. Following Xu et al. [93], we use a two-stage paradigm: (1) full-parameter fine-tuning on a monolingual corpus, and (2) LoRA fine-tuning on parallel data. We build on ALMA-7B, already stage-1 fine-tuned, which we fine-tune on the parallel data from Xu et al. [93] (See Apx E.8), comprising WMT'17–20 test sets and Flores-200 dev/test sets, restricted to English–German ($\sim$14k pairs). Evaluation uses the newstest2014 subset from Ott et al. [63], containing 500 English sentences with ten German references each.

Figure 4 displays the results, with a legend that mirrors the one in Figure 2. `LoRA-MCL` (circles) is trained with $K = 3$ and $\varepsilon = 0.05$, and each model generates 3 sequences per input at test time. We evaluate `LoRA-MLE` with ranks $r = 16$ (diamond) and $r = 48$ (square), using BS (blue) with widths 3, 6, and 9, and DBS (green) with $\lambda = 0.8$. We follow the quality–diversity evaluation protocol of Shen et al. [76]. Scores are reported using both Leave-One-Out BLEU (Loo-BLEU) and Pairwise-BLEU [76]. The results show that `LoRA-MCL` achieves a strong balance between quality and diversity, confirming the effectiveness of the method.

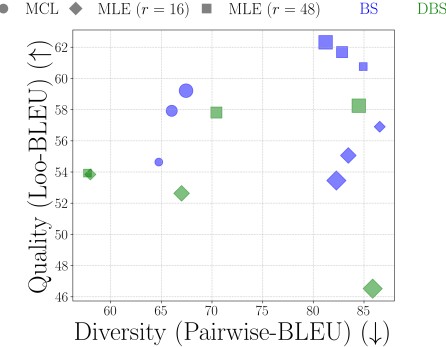

Figure 4: **Quality vs. Diversity in MT.**

## 6    RELATED WORK

**MCL to predict diverse and plausible outputs.** MCL [20, 38] is a training paradigm that minimizes the WTA loss across a set of models, encouraging specialization [73, 40]. While the collapse issue needs to be addressed [73, 66], MCL has demonstrated broad applicability across tasks [37, 74, 17], typically using a shared backbone and multiple heads. However, using multiple heads may be impractical in LLMs. Mixture-of-Experts (MoE) [29, 75] offers an alternative for managing computational costs, since only a subset of experts is active at each forward pass. In LLMs, however, MoE has primarily been used to improve scalability rather than to encourage diversity, and suffers from redundancy among experts [31]. While MoE can be adapted to the LoRA setting [91, 41], there is no clear consensus on the degree of specialization achieved. To our knowledge, this work is the first to adapt MCL to next-token language modeling using multiple LoRA modules.

**Generating Multiple Outputs with Language Models.** Language models are commonly trained via next-token prediction, framed as Maximum Likelihood Estimation. This is arguably the most popular method for training large-scale language models [77, 68, 81], including those that take audio or images as input [6, 43], with much of its success attributed to tokenization [50, 70]. Generating diverse and plausible sequences at inference remains challenging: $(i)$ Sampling methods [25, 15, 57] may be unreliable depending on the chosen parameters; $(ii)$ Exact MAP decoding is intractable due to the exponential search space [14]; $(iii)$ Strategies like Beam Search often yield repetitive or overly coherent outputs [15, 25, 33]. Diverse Beam Search [56] inject diversity through test-time parameters (penalty $\lambda$), but in contrast, `LoRA-MCL` infers diversity from the input's inherent ambiguity.

**Diversity in Audio and Visual Captioning.** Audio [11, 54, 55] and image captioning [2, 23, 26] have traditionally relied on MLE-trained, task-specific models. A key challenge is the limited diversity of generated captions, leading to training objectives tailored to this issue [56, 95, 94, 100, 85, 86, 48]. These approaches often require architectural changes and models trained from scratch. With the rise of general-purpose multimodal LLMs [6, 43], addressing the diversity–quality trade-off remains critical. We show that `LoRA-MCL` effectively tackles this at the fine-tuning stage.

## 7    CONCLUSION

LoRA-MCL a paradigm that combines MCL with LoRA to train language models for diverse, plausible predictions. Our analysis shows that when the target sequence is a mixture, LoRA-MCL can capture the modes of the data distribution. We validate this on Markov chains as well as the tasks of audio, image captioning, and machine translation. Future work will focus on interpreting the concepts learned by each model relative to input ambiguity, with applications to uncertainty estimation.

**Limitations.** In `LoRA-MCL` (Relaxed), setting $\varepsilon$ too high guarantees that all hypotheses receive gradients, but it may also overly homogenize the models and thus reduce diversity. A similar effect arises in the annealed variant, where the temperature scheduler parameters influence performance. Although MCL performs robustly across a broad range of parameter values, identifying the optimal configuration remains challenging. Finally, recent LoRA variants such as [21, 101] may provide an orthogonal direction of improvement for our method.

## 8 REPRODUCIBILITY STATEMENT

All the experimental settings are extensively described in the Appendices D and E to provide enough detail to reproduce the experiments. Our implementation leverages existing open source repositories.

We intend to publish our code and checkpoints upon paper acceptance.

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

CONTENTS

## A   NOTATIONS AND SETUP

In the following, $x \triangleq (x_t)_{t=1}^T \in \mathcal{V}^T$ be a sequence of $T$ tokens belonging to a finite vocabulary $\mathcal{V} = \{1, \ldots, |\mathcal{V}|\}$, and $c \triangleq (c_t)_{t=1}^\tau \in \mathcal{C}$ be a sequence of $\tau$ context of embeddings of dimension $d$. In the following $\mathcal{X} = \mathcal{V}^T$ and $\mathcal{C} = (\mathbb{R}^d)^\tau$.

Language modeling aims at learning the law $p(x \mid c) = \prod_{t=1}^T p(x_t \mid x_{<t}, c)$ using a model $p_\theta$ with parameters $\theta \in \Theta$, by maximum likelihood estimation, minimizing the negative log-likelihood loss

$$\mathcal{L}(\theta) = -\mathbb{E}_{c,x}[\log p_\theta(x \mid c)] = \mathbb{E}_{c,x}\left[-\sum_{t=1}^T \log p_\theta(x_t \mid x_{<t}, c)\right], \tag{10}$$

where $x_{<t}$ denotes the sequence of tokens prior to time $t$. In practice, we assume that $p_\theta = s_\eta \circ f_\theta$, where $f_\theta(x_{<t}, c) \in \mathbb{R}^{\mathcal{V}}$ are the predicted logits and $s_\eta : z \in \mathbb{R}^{\mathcal{V}} \mapsto \left( \frac{\exp(z_j/\eta)}{\sum_{q=1}^{|\mathcal{V}|} \exp(z_q/\eta)} \right) \in [0, 1]^{\mathcal{V}}$ is the $\mathrm{softmax}$ operator with temperature $\eta > 0$.

In the following, we make the following assumption.

**Assumption 1** (Expressiveness). *In the following, we assume that the model $p_\theta$ is perfectly expressive. Formally, let $\mathcal{F}_\theta \triangleq \{p_\theta : \mathcal{C} \to \mathcal{P}(\mathcal{X})\}$ be the family of conditional distributions realized by the model, where $\mathcal{P}(\mathcal{X})$ is the set of probability distributions on $\mathcal{X}$. We assume the family $\mathcal{F}_\theta$ is perfectly expressive, that is $\mathcal{F}_\theta = \{p^\star : \mathcal{C} \to \mathcal{P}(\mathcal{X})\}$.*

First, note that we have the Proposition 2, a well-known result that is due to the Gibbs inequality (e.g., [47]), for which we provide a proof for completeness.

**Proposition 2** ([47]). *Under Assumption 1, for the next-token prediction loss (10), one can show that*

$$\min_\theta \mathcal{L}(\theta) = \mathcal{H}(x \,|\, c) , \tag{11}$$

*where $\mathcal{H}(x \,|\, c) \triangleq -\mathbb{E}_{c,x}[\log p(x \,|\, c)]$ is the entropy of $p(\cdot \,|\, c)$.*

*Proof.* Let us denote $\mathscr{S}(p) = \{(x, c) \,|\, p(x, c) > 0\}$ the support of $p$. We use the convention $p(x, c) \log p(x, c) = 0$ for $(x, c) \in \mathcal{X} \times \mathcal{C} - \mathscr{S}(p)$. Because $\log$ is a concave function, we have under Jensen's inequality:

$$-\int_{\mathscr{S}(p)} \log \left[ \frac{p_\theta(x \,|\, c)}{p(x, c)} \right] p(x, c) \, \mathrm{d}x\mathrm{d}c \geq -\log \left( \int_{\mathscr{S}(p)} \frac{p_\theta(x \,|\, c)}{p(x, c)} p(x, c) \, \mathrm{d}x\mathrm{d}c \right) \geq 0 . \tag{12}$$

However, because of the convention, the left-hand side of (12) is also equal to the integral over $\mathcal{X} \times \mathcal{C}$. This shows:

$$-\int_{\mathcal{X} \times \mathcal{C}} p(x, c) \log p_\theta(x \,|\, c) \, \mathrm{d}x\mathrm{d}c \geq -\int_{\mathcal{X} \times \mathcal{C}} p(x, c) \log p(x \,|\, c) \, \mathrm{d}x\mathrm{d}c = \mathcal{H}(x \,|\, c) ,$$

where the equality is reached for parameter $\theta$ such that $p_\theta = p$, whose existence is guaranteed by Assumption 1. $\qquad\square$

In the following, we denote the Kullback–Leibler divergence between two distributions $\alpha$ and $\beta$ as $\mathrm{KL}(\alpha \,\|\, \beta) \triangleq \int_{\mathscr{S}(\beta)} \alpha(x) \log \frac{\alpha(x)}{\beta(x)} \mathrm{d}x$, where $\mathscr{S}(\beta) \triangleq \{x \in \mathcal{X} \,|\, \beta(x) > 0\}$. Note that we have the equality:

$$-\mathbb{E}_\alpha[\log \beta(x)] = \mathrm{KL}(\alpha \,\|\, \beta) + \mathcal{H}(\alpha) , \tag{13}$$

where the left-hand side is usually referred as the Cross-Entropy, and $\mathcal{H}(\alpha) \triangleq -\mathbb{E}_\alpha[\log \alpha(x)]$ is the entropy of $\alpha$. When the context is clear, we will also write the entropy of a distribution $\alpha$ as $\mathcal{H}(x)$ where $x \sim \alpha$.

## B    PROOF OF PROPOSITION 1

Let us now consider the following assumptions.

**Assumption 2** (Mixture of latent processes). *The data-generating process writes in form $p(x \,|\, c) = \sum_{k=1}^{K} p(z_k \,|\, c) \, p(x \,|\, z_k, c)$. The Mixture is said to be uniform if $\forall k, p(z_k \,|\, c) = \frac{1}{K}$.*

**Assumption 3** (Minimization of the true risk). *The batch size is large enough so that the minimization of the empirical risk comes down to minimizing the true risk (10).*

**Remark 1.** *Under Assumptions 2 and 3, the optimal reachable loss by maximum likelihood estimation is $\min_\theta \mathcal{L}(\theta) = \mathbb{E}_c [\mathcal{H}(x \,|\, c)]$, where $x \sim \frac{1}{K} \sum_{k=1}^{K} p(x \,|\, z_k, c)$.*

**Assumption 4** (Disjoint components). *This assumption states that $p(x \,|\, c, z_s) = 0$ when $p(x \,|\, c, z_k) > 0$, for $s \neq k$.*

**Proposition 3.** *Under Assumptions 1, 2, and 3, we have that:*

(i) *The Winner-Takes-All two-step optimization in* `LoRA-MCL` *acts as a conditional form of the hard-EM algorithm.*

(ii) *Under Assumption 4, and assuming (with one permutation) that* $p(x \mid z_k, c) = p(x \mid c; \theta_k)$ *for each* $k$, $\mathcal{L}^{\mathrm{WTA}}(\theta) = -\mathbb{E}_{x,c} \left[ \max_{k=1,\ldots,K} \log p(x \mid c, z_k) \right]$. *In this case, we also have:*

$$\mathcal{L}^{\mathrm{WTA}}(\theta) = \mathcal{H}(x \mid c, z) \triangleq \mathbb{E}_c \left[ \sum_{k=1}^{K} p(z_k \mid c) \mathcal{H}(x \mid c, z_k) \right] , \qquad (14)$$

*where* $\mathcal{H}(x \mid c, z)$ *is the conditional entropy given the random variable* $z$.

(iii) *We have the following inequalities:*

$$\min_\theta \mathcal{L}(\theta) - \log K \overset{(a)}{\leq} \min_\theta \mathcal{L}^{\mathrm{WTA}}(\theta) \overset{(b)}{\leq} \mathcal{H}(x \mid c, z) \overset{(c)}{\leq} \min_\theta \mathcal{L}(\theta) , \qquad (15)$$

*where* $\min_\theta \mathcal{L}(\theta) = \mathcal{H}(x \mid c)$.

*Proof of (i)* First, let us remind that the hard-EM consists of fitting a distribution $p_\theta(x, z)$ to observed data $x \sim p(x)$ where $z$ are (unknown) hidden variables. The fitting starts from randomly initialized parameters $\theta$ and latent variables $z$. It consists of repeating the following operations at each iteration $t$ until convergence:

1. (Expectation) $z_k^\star = \mathrm{argmax}_k \, p(x, z_k; \theta^{(t)})$
2. (Maximization) $\theta^{(t+1)} = \mathrm{argmax}_\theta \, p(x, z_k^\star; \theta^{(t)})$

Let us define:

$$D(\theta, q) \triangleq \int_\mathcal{X} \sum_{k=1}^{K} q(k \mid x) \log p(x, z_k; \theta) \, \mathrm{d}p(x), \quad D(\theta) \triangleq \int_\mathcal{X} \max_{k=1,\ldots,K} \log p(x, z_k; \theta) \, \mathrm{d}p(x) , \quad (16)$$

where $q$ is a discrete distribution over $\{1, \ldots, K\}$ with exactly one non-zero component that controls the assignment of each $x$ to a fixed $k^\star$. Let us define $q(\theta)$ as the discrete distribution defined so that $q(k \mid x; \theta) \triangleq \mathbf{1}[k = \mathrm{argmax}_s \, p(x, z_s; \theta)]$. Note that then $D(\theta) = D(\theta, q(\theta))$.

For the vanilla (or soft) EM algorithm, the complete data log-likelihood $\int_\mathcal{X} \log p(x; \theta) p(x) \mathrm{d}x$ is expected to increase at each iteration $t$. Similarly, for the hard-EM, we have that the $D(\theta)$ increases at each iteration.

Indeed, the expectation step comes down to computing $q(\theta^{(t)})$. For for the Maximization step, we have: $D\left(\theta^{(t+1)}, q(\theta^{(t)})\right) \geq D\left(\theta^{(t)}, q(\theta^{(t)})\right)$, by definition. At the next expectation step, we have: $D\left(\theta^{(t+1)}, q(\theta^{(t+1)})\right) \geq D\left(\theta^{(t+1)}, q(\theta^{(t)})\right)$, because $q(\theta^{(t+1)})$ computes the best assignment given the parameters $\theta^{(t+1)}$. This shows that $D(\theta^{(t+1)}) \geq D(\theta^{(t)})$.

First note that the main difference compared to the vanilla form of the (hard) EM algorithm, is that the goal is to fit here a *conditional* distribution $p(x \mid c)$ given pair $(c, x) \sim p(c, x)$. Furthermore, step 2 performs a gradient step (of the neural network weights) instead of a full maximization.

Note that under Assumption 2 the complete data log-likelihood writes as $\log p(x \mid c; \theta) = \log \left[ \sum_{k=1}^{K} p(x, z_k \mid c; \theta) \right]$. However, the variables $z_k$ are not known in practice. `Lora-MCL` works by analogy with the Hard-EM algorithm, which consists, in the Expectation step, of picking for each pair $(x; c)$, $k^\star(x, c) = \mathrm{argmax}_k \, p(x \mid c; \theta_k)$. Indeed, under Assumption 3, each training step of `Lora-MCL` writes as the optimization of

$$\mathcal{L}^{\mathrm{WTA}}(\theta) = -\int_{\mathcal{X} \times \mathcal{C}} \max_{k=1,\ldots,K} \log p(x \mid c; \theta_k) \mathrm{d}p(c, x) , \qquad (17)$$

which is expected to decrease at each training iteration. Note that because the loss is bounded from below (by 0), the sequence of real numbers $\{\mathcal{L}^{\mathrm{WTA}}(\theta^{(t)})\}_{t \geq 0}$ is therefore expected to converge.

To conclude, we can view $(\theta_1, \ldots, \theta_K)$ as the parameters involved in the estimation of the modes of the conditional distribution with $p(x \mid c) = \sum_{k=1}^{K} p(x \mid c; \theta_k) p(\theta_k \mid c)$. Note that the current form of the algorithm does not estimate the weight of each mode $p(\theta_k \mid c)$, further work could include incorporating *scoring* heads to estimate $p(\theta_k \mid c)$ each $k$ as in Letzelter et al. [39]. □

We then expect to be able to recover the distributions (with one permutation) $\{p(x \mid c, z_k)\}$ from estimated $\{p(x \mid c; \theta_k)\}$, assuming identifiability of the data generating mixture, which we expect to be made easier if the components are enough separated (See e.g., [71, Par. 2.5] or [53, Sec. 2.2]).

*Proof of (ii)* Let us assume that (with one permutation) $p(x \mid z_k, c) = p(x \mid c; \theta_k)$ for $k \in \{1, \ldots, K\}$. This is possible thanks to Assumption 1. Let us show that (14).

Let us define:
$$\mathcal{X}_k(c, \theta) \triangleq \left\{ x \in \mathcal{X} \mid \log p(x \mid c, \theta_k) \geq \log p(x \mid c, \theta_s) \ \forall s \in \{1, \ldots, K\} \right\}. \qquad (18)$$

In this case, the WTA loss (2) writes as

$$\mathcal{L}^{\mathrm{WTA}}(\theta) = - \int_{\mathcal{C}} \sum_{k=1}^{K} \int_{\mathcal{X}_k(c,\theta)} \log p(x \mid c; \theta_k) \, p(x \mid c) \, \mathrm{d}x \, p(c) \mathrm{d}c$$

$$= - \int_{\mathcal{C}} \sum_{k=1}^{K} \sum_{s=1}^{K} \int_{\mathcal{X}_k(c,\theta)} \log p(x \mid c; z_k) \, p(x \mid c; z_s) \, p(z_s \mid c) \, \mathrm{d}x \, p(c) \, \mathrm{d}c$$

$$= - \int_{\mathcal{C}} \sum_{k=1}^{K} \int_{\mathcal{X}_k(c,\theta)} \log p(x \mid c; z_k) \, p(x \mid c; z_k) \, p(z_k \mid c) \, \mathrm{d}x \, p(c) \, \mathrm{d}c \quad \text{by Asm. 4}$$

$$= - \int_{\mathcal{C}} \sum_{k=1}^{K} \mathcal{H}(x \mid c; z_k) \, p(z_k \mid c) \, p(c) \mathrm{d}x \, \mathrm{d}c \quad \text{as} \int_{\mathcal{X}_k(c,\theta)} p(x \mid c; z_k) \mathrm{d}x = 1.$$

$$= \mathbb{E}_c \left[ \sum_{k=1}^{K} p(z_k \mid c) \mathcal{H}(x \mid c; z_k) \right].$$

□

*Proof of (iii)* Let us show that:
$$\min_\theta \mathcal{L}(\theta) - \log K \overset{(a)}{\leq} \min_\theta \mathcal{L}^{\mathrm{WTA}}(\theta) \overset{(b)}{\leq} \mathcal{H}(x \mid c, z) \overset{(c)}{\leq} \min_\theta \mathcal{L}(\theta).$$

$(a)$: First, we have $\max_{k=1,\ldots,K} p(x \mid c, z_k) \leq \sum_{k=1}^{K} p(x \mid c, \theta_k)$. Therefore,

$$\mathcal{L}^{\mathrm{WTA}}(\theta) \geq - \mathbb{E}_{x,c} \left[ \log \frac{1}{K} \sum_{k=1}^{K} p(x \mid c, \theta_k) \right] - \log K$$

$$= \underbrace{\mathrm{KL} \left[ p(x \mid c) \, \| \, \frac{1}{K} \sum_{k=1}^{K} p(x \mid c, \theta_k) \right]}_{\geq 0} + \mathcal{H}(x \mid c) - \log K \quad \text{by (13)}$$

$$\geq \mathcal{H}(x \mid c) - \log K.$$

Because $\min_\theta \mathcal{L}(\theta) = \mathcal{H}(x \mid c)$, we have shown that $(a)$ occurs when the KL term vanishes, which is when $p(x \mid c) = \frac{1}{K} \sum_{k=1}^{K} p(x \mid c, \theta_k)$.

$(b)$: We have
$$\mathcal{L}^{\mathrm{WTA}}(\theta) = - \mathbb{E}_x \left[ \max_{k=1,\ldots,K} \log p(x \mid c, \theta_k) \right]$$

$$= - \mathbb{E}_z \mathbb{E}_{x \mid z} \left[ \log \frac{\max_{k=1,\ldots,K} p(x \mid c, \theta_k)}{p(x \mid c, z)} \right] \underbrace{- \mathbb{E}_z \mathbb{E}_{x \mid z} \left[ \log p(x \mid c, z) \right]}_{\mathcal{H}(x \mid c, z)}.$$

Now let us leverage Assumption 1 to choose $\tilde{\theta}_k$ such that $p(x \mid c, \tilde{\theta}_k) = p(x \mid c, z_k)$ for each $k \in \{1, \ldots, K\}$. In this case, $\max_k p(x \mid c, \tilde{\theta}_k) \geq p(x \mid c, z)$ for each $z \in \{z_1, \ldots, z_K\}$, and $-\mathbb{E}_z \mathbb{E}_{x \mid z} \left[ \log \frac{\max_k p(x \mid c, \tilde{\theta}_k)}{p(x \mid c, z)} \right] \leq 0$. Then,

$$\min_\theta \mathcal{L}^{\mathrm{WTA}}(\theta) \leq \mathcal{L}(\tilde{\theta}_1, \ldots, \tilde{\theta}_K) \leq \mathcal{H}(x \mid c, z) \,,$$

which proves $(b)$.

Finally $(c)$ can be directly deduced from the inequality $\mathcal{H}(x \mid c, z) \leq \mathcal{H}(x \mid c)$. $\qquad \square$

## C    PROOF OF COROLLARY 1

Let us consider the following assumptions.

**Assumption 5** (Markov Chain). *We assume that the data-generating process writes as a uniform mixture of Markov chains of order $n \in \mathbb{N} \setminus \{0\}$, that is, for each $t$ and each $k$, $p(x_t \mid x_{<t}, c, z_k) = p(x_t \mid x_{t-1}, \ldots, x_{t-n}, c, z_k)$.*

**Corollary 2.** *As per Assumption 5, let us assume that the data-generating process writes as a uniform mixture of Markov chains of order $n = 1$. Let $\hat{P}(\theta) \triangleq (p(x_{t+1} = j \mid x_t = i))_{i,j}$ be the predicted transition matrix when using a language model with parameters $\theta$. Under the same assumptions that in Proposition 1, we have:*

*(i) Whenever the maximum likelihood estimator trained with next-token-prediction (10) reaches its optimal loss $\mathcal{L}(\theta)$, we have*

$$\hat{P}(\theta)_{i,j} = \sum_{k=1}^K p(z = z_k \mid x_t = i)(P_k)_{i,j} = \frac{1}{\sum_{s=1}^K (\pi_s)_i} \sum_{k=1}^K (\pi_k)_i (P_k)_{i,j} \,,$$

*where $\pi_k \in [0,1]^\mathcal{V}$ is the stationary distribution of $P_k$.*

*(ii) The inequality (7) holds in this context, where the conditional entropy $\mathcal{H}(x \mid z)$ can be computed by a weighted sum the entropy rate of each of the $K$ Markov Chains:*

$$\mathcal{H}(x \mid z) = -T \sum_{k=1}^K \sum_{i=1}^{|\mathcal{V}|} (\pi_k)_i \Big[ \sum_{j=1}^{|\mathcal{V}|} (P_k)_{i,j} \log(P_k)_{i,j} \Big]. \tag{19}$$

*The entropy $\mathcal{H}(x)$, which is the lower bound of the MLE baseline, can be computed either exactly for short sequences, or approximated, e.g., through Monte-Carlo integration.*

*Proof of (i)* The Cross entropy of the maximum-likelihood model is optimal whenever $p_\theta = p$.

In this case, for each $i, j \in \{1, \ldots, |\mathcal{V}|\}$, we have $p_\theta(x_{t+1} = j \mid x_t = i) = p(x_{t+1} = j \mid x_t = i)$ and

$$p_\theta(x_{t+1} = j \mid x_t = i) = \sum_{k=1}^K p(z = z_k \mid x_t = i)\, p(x_{t+1} = j \mid x_t = i, z_k)\,. \tag{20}$$

Furthermore, by Bayes' rule, we have:

$$p(z = z_k \mid x_t = i) = \frac{p(x_t = i \mid z = z_k) p(z = z_k)}{\sum_{s=1}^K p(x_t = i \mid z = z_s) p(z = z_s)} = \frac{(\pi_k)_i}{\sum_{s=1}^K (\pi_s)_i} \,,$$

because we have assumed a uniform prior over the mixture components, i.e., $p(z = z_k) = \frac{1}{K}$, and we assumed stationary regime so that $p(x_t = i \mid z = z_k) = (\pi_k)_i$. $\qquad \square$

*Proof of (ii)* Because we assumed first-order Markov Chains, we have

$$\mathcal{H}(x) = \sum_{t=1}^T \mathcal{H}(x_t \mid x_{<t})\,.$$

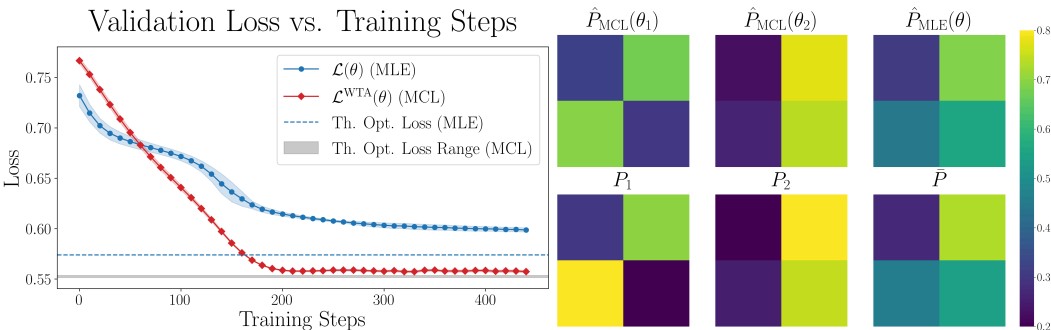

Figure 5: **Comparison of `LoRA-MCL` with standard maximum likelihood estimation (MLE).** The setup mirrors that of Figure 1, but uses different transition matrices. While the overall behavior remains consistent with Figure 1, we observe a distinct transition in the MLE loss. We interpret this as evidence that MLE increasingly incorporates contextual information as training progresses (see also [50]).

Because we assumed stationary Markov chains, we have that $\mathcal{H}(x_t \mid x_{t-1})$ doesn't depend on $t$ and $\mathcal{H}(x_t \mid x_{t-1}) = \mathcal{H}(x_2 \mid x_1) = -\sum_{i=1}^{|\mathcal{V}|}(\pi_k)_i \Big[\sum_{j=1}^{|\mathcal{V}|}(P_k)_{i,j}\log(P_k)_{i,j}\Big]$ (see [8] page 66, Theorem 4.2.4). $\qquad\square$

Note that we considered first-order Markov chains in our analysis. However we expect the properties to generalize to higher orders ($n > 1$) by using transition matrices in the form $P \in [0,1]^{\mathcal{V}^{n+1}}$ where

$$P_{i_1,\ldots,i_{n+1}} = p(x_{t+1} = i_{n+1} \mid x_t = i_n, \ldots, x_{t-n+1} = i_1) \,.$$

# D  EXPERIMENTAL DETAILS OF SECTION 4.3

The results in Figure 1 were obtained using the setup described in this section. The same illustration with different transition matrices is shown in Figure 5. Note that in both figures, the loss and theoretical quantities are normalized by $T - 1$, since the first token is excluded from the computation.

**Dataset.** We used a uniform mixture of two first-order homogeneous Markov chains with transition matrices $(P_1, P_2) = \big(P(p_1, q_1), P(p_2, q_2)\big)$, with $P(p,q) \triangleq \begin{bmatrix} 1-p & p \\ q & 1-q \end{bmatrix}$ and $p, q \in [0,1]$. For the experiment in Figure 1, we used $(p_1, q_1) = (0.2, 0.9)$ and $(p_2, q_2) = (0.8, 0.25)$. Figure 5 uses $(p_1, q_1) = (0.7, 0.8)$ and $(p_2, q_2) = (0.8, 0.25)$. The first state of the sequences sampled according to the stationary distribution of the Markov chain given by $\pi_k = \frac{1}{p_k + q_k}(q_k, p_k)$ (see e.g., [50]). The sequences have a fixed length of $T = 32$.

**Architecture.** We considered a GPT-2-like architecture [68] using the GPT-Neo implementation [4] using local-attention suggested by Makkuva et al. [50] to improve convergence on Markov chain data (with window size of 5). The model has a hidden size of 64, 2 layers of transformer blocks with 2-heads attention. LoRA adapters for the MLE baseline have rank $r = 64$, $\alpha = 64$, and dropout disabled. To align the numbers of parameters when $K = 2$, we used $r = 32$ (and $\alpha = 32$) for `LoRA-MCL`. The models have a total of 65,536 trainable parameters over a total of 230,912. Note that aligning the ranks of `LoRA-MCL` and `LoRA-MCL` leads to the same conclusions. Weights of the base model were kept frozen (both for `LoRA-MLE` and `LoRA-MCL`) to mimic the dynamics on larger language models.

**Training details.** We used a cosine scheduler with learning rate of $1e-4$, weight decay of $1e-3$ with AdamW optimizer, with $(\beta_1, \beta_2) = (0.9, 0.95)$ as in Makkuva et al. [50]. We used a batch size of 128, and trained for 500 iterations, with validation loss computed every 10 steps. Only the first 450 iterations are plotted in the Figures 1 and 5. For the `LoRA-MCL` runs, we trained with vanilla Winner-Takes-All update. Figure 1 (left) shows the mean and standard deviation of the Winner-Takes-All validation loss across three training seeds, for $K = 1$ (`LoRA-MLE`) and $K = 2$ (`LoRA-MCL`).

**Theoretical quantities computations** To verify the theoretical results, we computed the following quantities:

- **Theoretical Optimal Loss of the MLE model.** It is expressed as $\mathcal{H}(x)$ where $x \sim \frac{1}{K} \sum_{k=1}^{K} p(x \mid z_k)$. It can be approximated by Monte-Carlo sampling with samples $(z_s, x_s) \sim p(x, z)$ by first sampling modes $z_s \sim \mathcal{U}\{1, \ldots, K\}$, then $x_s \mid z_s \sim p(x \mid z_s)$, and by computing

$$-\frac{1}{N} \sum_{s=1}^{N} \sum_{t=2}^{T} \log{(P_s)_{x_{s,t}, x_{s,t+1}}} . \tag{21}$$

  Here, $(P_s)_{x_{s,t}, x_{s,t+1}}$ denotes the entry of $P_s$ at row $x_{s,t}$ and column $x_{s,t+1}$, and $N$ is the total number of samples. We used $N = 50{,}000$ here. Note that the term corresponding to $t = 1$ in (21) is discarded, as the first token is typically excluded from the loss computation. For the illustration, (21) was normalized by $T - 1$, since the first token is discarded.

- **Theoretical Optimal Loss of the MCL model.** It is computed by subtracting $\log K$ to the Theoretical optimal loss (21) of the MLE model. To verify $(iii)$ in Proposition 1, we also computed the mixture of entropy rates given by (19), using a normalization factor of $T - 1$.

# E  EXPERIMENTAL DETAILS CAPTIONING AND TRANSLATION TASKS

## E.1  SETUP

Dataset statistics are provided in Table 4.

**Audio Datasets.** We conducted experiments on two audio captioning datasets: Clotho-V2 [16, 12] and AudioCaps [18, 34]. For both datasets, we used the official training, validation, and test splits. Clotho-V2 provides five reference captions per audio clip across all splits, whereas AudioCaps includes a single caption per clip in the training set and five captions per clip in the validation and test sets. During training on Clotho-V2, which is performed on 10 epochs, at each epoch, one of the five reference captions is sampled uniformly at random for each audio clip.

**Vision datasets.** We conducted experiments on the TextCaps dataset. Specifically, we used the official training split for training and the official validation split as our test set. Since each image in the dataset is annotated with five different captions, we duplicated each image five times—associating each duplicate with a distinct caption—to ensure that all reference captions are seen during a single training epoch.

**Preprocessing of audio data.** Following the implementation of [35], for Clotho and AudioCaps raw audio files are resampled from $44.1\,\text{kHz}$ and $32\,\text{kHz}$ respectively to $16\,\text{kHz}$. They are then cropped to a maximum length of 30 seconds for Clotho and 10 seconds for AudioCaps. The data is then fed to the Qwen-2-Audio pipeline, which includes conversion of the raw waveform into a 128-channel mel-spectrogram, with a window size of 25 ms and a hop size of 10 ms. As input to the language model, we provide the constant prompt "`Generate the caption in English:`", following the documentation of Qwen-2-Audio in the transformers library.

**Preprocessing of image data.** Our image pre-processing pipeline follows the recipe of LLaVA (i.e., resizing to $(336, 336)$ and normalization using CLIP mean and standard deviation).

For both modalities, we used the HuggingFace transformers [90] and PEFT [52] Python libraries as part of the implementation. Note that for each of the experiments, we set the repetition penalty [33] to 1.1 for decoding.

## E.2  METRICS

In the following, we describe how to assess a language model in generating sequences $\hat{x}^1, \ldots, \hat{x}^K$ conditioned on a context $c$ (for instance, an audio recording paired with a captioning prompt in the case of Audio Captioning) using a given decoding method. In the case of `LoRA-MCL`, we denote by $\theta_1, \ldots, \theta_K$ the parameters of the language models corresponding to each hypothesis. We assume access to a set of $R \geq 1$ references $x^1, \ldots, x^R$ for each context, which can be regarded as samples

Table 4: **Statistics of the audio and image captioning datasets.** Num. samples includes {train, validation, test} sets, except for AudioCaps, where we used only {train, validation} sets.

| Dataset | Num. samples | Duration (h) | Num. Captions | Modality |
|---|---|---|---|---|
| AudioCaps [34] | 48,286 | 134.1 | 54 K | Audio |
| Clotho [12] | 5,929 | 37.0 | 30 K | Audio |
| TextCaps [78] | 25,119 | N/A | 126 K | Image |

from the *ground-truth* conditional distribution $p(x \mid c)$. The evaluation is performed on a dataset of $N$ pairs $(c_i, \{x_i^1, \ldots, x_i^R\})_{i=1}^N$.

### E.2.1 NEGATIVE LOG-LIKELIHOOD

**Test NLL and Perplexity ($\downarrow$).** A standard way to evaluate a language model is through its test loss (e.g., [92]), which measures the average likelihood of the reference sentences under the trained model. When considering multiple language models, the oracle NLL is defined by averaging, for each reference, the best NLL across the $K$ hypotheses:

$$\mathrm{NLL} \triangleq -\frac{1}{NR} \sum_{i=1}^N \sum_{x \in \{x_i^1, \ldots, x_i^R\}} \max_{k=1,\ldots,K} \frac{1}{T(x)} \sum_{t=1}^{T(x)} \log p(x_t \mid x_{<t}, c_i, \theta_k) , \tag{22}$$

where $T(x)$ denotes the length of the reference sequence $x$ (in number of tokens), and $\log$ refers to the natural logarithm. In the context of LLMs, perplexity [30] is usually defined as $\mathrm{PPL} = \exp(\mathrm{NLL})$. It can be interpreted as the effective number of equally likely tokens among which the model is uncertain when predicting the next token. In particular, if the model always predicts a uniform distribution over the vocabulary $\mathcal{V}$ then $\mathrm{PPL} = |\mathcal{V}|$.

### E.2.2 NATURAL LANGUAGE GENERATION QUALITY METRICS

Originally developed within the human translation community, BLEU (Bilingual Evaluation Understudy) [64], ROUGE (Recall-Oriented Understudy for Gisting Evaluation) [42], and METEOR (Metric for Evaluation of Translation with Explicit Ordering) [3] were introduced to measure the closeness between a machine translation and a professional human translation. For consistency, all sequences are tokenized using the Penn Treebank Tokenizer (PTB Tokenizer). Implementations rely on the AAC Metrics and COCO Caption libraries. An $n$-gram refers to a group of $n$ consecutive tokens in a tokenized sequence. When comparing candidate and reference sentences, we use the $F_\beta$ score, which balances precision and recall:

$$F_\beta(\mathrm{precision}, \mathrm{recall}) \triangleq \frac{(1+\beta^2)\mathrm{precision} \cdot \mathrm{recall}}{\beta^2 \mathrm{precision} + \mathrm{recall}} . \tag{23}$$

Each metric $\mathcal{M}$ is defined as a function of a candidate $\hat{x}^k$ and a set of references $X_i = \{x_i^1, \ldots, x_i^R\}$. However, these metrics do not natively handle multiple candidates (see e.g., [38, Sec.,4.3]; [36]). We therefore adopt a sentence-based oracle evaluation, where the final score is computed as

$$\mathcal{M}_{\mathrm{Oracle}} = \frac{1}{N} \sum_{i=1}^N \max_{k=1,\ldots,K} \mathcal{M}(\hat{x}_i^k, X_i) . \tag{24}$$

**BLEU ($\uparrow$).** For a given $n$, the modified $n$-gram precision $p_n$ (see Section 2.1 in Papineni et al. [64]), measures the fraction of the unigrams in a predicted caption that also appear in the reference captions, while clipping the numerator by the maximum number of times a word occurs in any single reference translation. $\mathrm{BLEU}_n$ combines the n-grams precision up to length $n$ computing a geometric mean of $p_s$ for $s = 1, \ldots, n$. It also applies a multiplicative brevity penalty BP to penalty too short predicted captions compared to the reference sequences lengths. In this case, we have $\mathcal{M} = \mathrm{BLEU}_n$ and:

$$\mathrm{BLEU}_n(\hat{x}, X) \triangleq \mathrm{BP}(\hat{x}, X) \times \left( \prod_{s=1}^n p_s(\hat{x}, X) \right)^{\frac{1}{n}} , \tag{25}$$

where $\mathrm{BP}(\cdot, \cdot)$ and $p_s(\cdot, \cdot)$ are function of the candidates and the set of references (please refer to Section 2.3 of [64], and the documentation for details).

**ROUGE** ($\uparrow$). The ROUGE family of metrics [42] was originally introduced for the automatic evaluation of summaries, following BLEU, and is based on measuring $n$-gram co-occurrence between candidate and reference sequences. In this work, we use $\mathrm{ROUGE}_L$, which relies on the Longest Common Subsequence (LCS) between a candidate and a reference. Formally, we set $\mathcal{M} = \mathrm{ROUGE}_L$ with

$$\mathrm{ROUGE}_L(\hat{x}, X) \triangleq \max_{x^r \in X} F_\beta(P_{\mathrm{LCS}}(\hat{x}, x^r), R_{\mathrm{LCS}}(\hat{x}, x^r)) \,, \tag{26}$$

where the precision and recall are defined as:

$$P_{\mathrm{LCS}}(\hat{x}, x^r) = \frac{\mathrm{LCS}(\hat{x}, x^r)}{\mathrm{length}(\hat{x})}, \ R_{\mathrm{LCS}}(\hat{x}, x^r) = \frac{\mathrm{LCS}(\hat{x}, x^r)}{\mathrm{length}(x^r)} \,,$$

where $\beta = 1.2$ by default.

**METEOR** ($\uparrow$). Unlike BLEU, METEOR [3] explicitly incorporates recall, measures word-level matches between a candidate and the reference, and accounts for grammaticality through word order. Here we set $\mathcal{M} = \mathrm{METEOR}$, defined as an $F_\beta$ score between precision and recall, with an additional fragmentation penalty that lowers the score when matching words are not in the correct order:

$$\mathrm{METEOR}(\hat{x}, X) \triangleq \max_{x^r \in X} (1 - \mathrm{Penalty}(\hat{x}, x^r)) \, F_\beta(P(\hat{x}, x^r), R(\hat{x}, x^r)) \tag{27}$$

where the precision are recall of the unigram matches between a candidate and a reference are given by

$$P(\hat{x}, x^r) = \frac{\mathrm{matches}(\hat{x}, x^r)}{\mathrm{length}(\hat{x})}, \ R(\hat{x}, x^r) = \frac{\mathrm{matches}(\hat{x}, x^r)}{\mathrm{length}(x^r)} \,,$$

with $\beta = \frac{1}{3}$ by default. The penalty depends on the number of chunks, i.e., groups of consecutive matches in the correct order, and penalizes disordered matches (See Section 2.2 of [3]).

### E.2.3 CAPTIONING EVALUATION.

Introduced specifically for image captioning, the more recent metrics CIDEr (Consensus-based Image Description Evaluation) [83], SPICE (Semantic Propositional Image Caption Evaluation) [1], and SPIDEr [44] have demonstrated stronger correlation with human judgment. As in the previous section, we employ sentence-based oracle evaluation as defined in (24).

**CIDEr** ($\uparrow$). CIDEr [83] assigns weights to $n$-grams in the candidate and reference captions using TF–IDF. An $n$-gram receives higher weight if ($i$) its term frequency (TF) is high, i.e., it appears often in the sequence, and ($ii$) it is informative, i.e., it occurs infrequently across the set of reference captions in the corpus. For each sequence $x$, we construct a vector $g^n(x)$ of dimension equal to the number of $n$-grams of length $n$, where the $k$-th component $[g^n(x)]_k$ is the TF–IDF weight of the $k$-th $n$-gram in $x$. CIDEr then computes the average cosine similarity between the candidate and each reference:

$$\mathrm{CIDEr}_n(\hat{x}, X) \triangleq \frac{1}{|X|} \sum_{x^r \in X} \frac{g^n(\hat{x}) \cdot g^n(X)}{\|g^n(\hat{x})\| \, \|g^n(X)\|} \,, \quad \mathrm{CIDEr} \triangleq \frac{1}{4} \sum_{n=1}^{4} \mathrm{CIDEr}_n \,, \tag{28}$$

where $\cdot$ denotes the euclidean dot product.

**SPICE** ($\uparrow$). SPICE [1] was designed to capture semantic adequacy by focusing on objects, attributes, and relations rather than surface $n$-gram overlap. Given a set of object classes $C$, relations $R$, and attributes $A$, each caption $x$ is parsed into a scene graph $T(x) = O(x) \cup E(x) \cup K(x)$ where $O(x) \subseteq C$ is the set of objects, $E(x) \subseteq O(x) \times R \times O(x)$ encodes relations between objects, and $K(x) \subseteq O(x) \times A$ represents attributes associated with objects. SPICE is defined as an $F_1$ score over the tuples in the semantic graphs:

$$\mathrm{SPICE}(\hat{x}, X) \triangleq F_1(P(\hat{x}, X), R(\hat{x}, X)) \,, \tag{29}$$

with precision and recall given by

$$P(\hat{x}, X) = \frac{\mathrm{matches}(T(\hat{x}), T(X))}{|T(\hat{x})|} \quad R(\hat{x}, X) = \frac{\mathrm{matches}(T(\hat{x}), T(X))}{|T(X)|} \,,$$

Here, $\mathrm{matches}(T(\hat{x}), T(X))$ counts the number of matching tuples between the candidate and the reference semantic graphs.

**SPIDEr** ($\uparrow$). SPIDEr [44] combines the strengths of CIDEr (capturing consensus through $n$-gram overlap) and SPICE (capturing semantic adequacy through scene graphs). It is defined as the simple average of the two metrics:

$$\mathrm{SPIDEr}(\hat{x}, X) \triangleq \frac{\mathrm{CIDEr}(\hat{x}, X) + \mathrm{SPICE}(\hat{x}, X)}{2} \; . \tag{30}$$

**sBERT Similarity** ($\uparrow$). BERTScore [99] leverages contextual embeddings from a pretrained BERT model [9] to compute token-level similarity. This allows it to ($i$) better match paraphrases and ($ii$) capture long-range dependencies while penalizing semantic changes. sBERT Similarity [105] considers Sentence BERT [72] (by default paraphrase-TinyBERT-L6-v2) as the pretrained model due to its capability so compare semantics with a single embedding per sequence. Denote their contextual embeddings by $e(x), e(\hat{x}) \in \mathbb{R}^d$. For multiple references, it is defined as an average cosine similarity:

$$\mathrm{sBERT}(\hat{x}, X) \triangleq \frac{1}{R} \sum_{r=1}^{R} \frac{e(\hat{x})^\top e(\hat{x}_r)}{\|e(\hat{x})\| \, \|e(x_r)\|} \; . \tag{31}$$

### E.2.4 DIVERSITY EVALUATION

Quality evaluation measures how well candidate captions match the references. By contrast, diversity evaluation considers only the set of generated candidates, irrespective of the references. Below we present the diversity metrics used in our setup. Note that the oracle formulation in (24) does not apply here.

**Div-$n$** ($\uparrow$). Div-$n$ is defined as the ratio between the number of distinct $n$-grams in the $K$ generated captions $\hat{x}^1, \ldots, \hat{x}^K$ and the total number of $n$-grams across those captions. Higher values indicate greater lexical diversity.

**mBLEU-$n$** ($\downarrow$). Mutual BLEU (mBLEU-$n$) is computed by treating each generated caption $\hat{x}^k$ as a candidate and evaluating its BLEU score against the remaining captions $\{x^s \mid s \neq k\}$. The final score is the average across all $K$ captions:

$$\mathrm{mBLEU}_n(\hat{x}^1, \ldots, \hat{x}^K) = \frac{1}{K} \sum_{k=1}^{K} \mathrm{BLEU}_n(\hat{x}^k, \{\hat{x}^s \mid s \neq k\}) \; . \tag{32}$$

Lower $\mathrm{mBLEU}_n$ values indicate greater diversity among the generated captions.

### E.3 TRAINING METHODS

We describe after specificity of each of the used training methods.

**LoRA-MLE.** Through the article, we refer to LoRA-MLE as the training method that optimizes (1), where the LoRA adapters are trained and the rest of the model is frozen. This corresponds exactly to the case where $K = 1$ is LoRA-MCL (as described hereafter). We used the default initialization of Low-Rank adapters in PEFT library, that In Low-Rank adapters, $A$ is initialized with Kaiming Uniform [22] initialization $A \sim \mathcal{U}[-\frac{1}{\sqrt{d}}, \frac{1}{\sqrt{d}}]$ where $d$ is the number of input features and $B$ is initialized with zeros.

**LoRA-MoE.** We denote by LoRA-MoE the use of multiple adapters within LoRA modules at each layer where LoRA is applied, trained in the style of a Mixture of Experts (MoE) [29, 75]. Let the hidden state be $\mathbf{x} \in \mathbb{R}^{b \times \mathcal{T} \times d}$, where $b$ is the batch size, $\mathcal{T}$ the sequence length, and $d$ the feature dimension. Under the *soft* MoE formulation of [59], the computation at layer $\ell$ is given by

$$\mathbf{x} \leftarrow f_\theta^\ell(\mathbf{x}) + \sum_k [\gamma(\mathbf{x})]_k B_\ell^k A_\ell^k \mathbf{x} \; ,$$

where $f_\theta^\ell$ is the frozen base model at layer $\ell$, and the *router* $\gamma : \mathbb{R}^d \to \Delta^{K-1}$ is applied independently to each token embedding $\mathbf{x}_{b,t} \in \mathbb{R}^d$, producing a $K$-dimensional vector of mixing weights. In

practice, we implement $\gamma$ as a linear projection followed by a softmax. In large language models, MoE typically employs hard (discrete) routing, e.g., top-$k$ selection from the router to control computational cost. In the context of LoRA, however, this tradeoff is less critical since LoRA computations are lightweight relative to the base model. Here, we adopt *soft* routing, which keeps the expert block fully differentiable and avoids reliance on gradient approximation or additional load-balancing losses [91, 41].

**LoRA-MCL.** This is the method introduced in this paper. At each LoRA-enabled layer $\ell$, a family of $K$ LoRA adapters $(A_k^\ell, B_k^\ell)_{k=1}^K$ is trained, while the rest of the model remains frozen. The training objective is

$$\mathcal{L}^{\text{WTA}}(\theta) = -\mathbb{E}_{c,x}\Big[\sum_{k=1}^K q_k(x,c) \log p(x \,|\, c; \theta_k)\Big],$$

where the coefficients $q_k$ depend on the chosen WTA mode. Let $k^\star(x,c) = \operatorname{argmax}_k p(x \,|\, c; \theta_k)$ be the index of the winning hypothesis for input $c$ and target $x$. In the *vanilla WTA* mode, we set $q_k(x,c) = \mathbf{1}[k = k^\star(x,c)]$, which directly optimizes the Oracle NLL Loss (22). However, this formulation risks *collapse*, where some hypotheses are rarely selected and thus under-trained. To mitigate this, Rupprecht et al. [73] proposed the *relaxed WTA* mode:

$$q_k(x,c) = (1 - \varepsilon)\mathbf{1}\left[k = k^\star(x,c)\right] + \frac{\varepsilon}{K-1}\mathbf{1}\left[k \neq k^\star(x,c)\right].$$

which gives higher weight to the winning hypothesis while still providing a small gradient to the others, controlled by $\varepsilon > 0$. Subsequent methods extend this idea by making $q_k$ a function of the training step $t$, thereby adjusting the contribution of non-winning hypotheses during learning [49, 60, 61, 66]. For example, in the *annealed MCL* method [66], a temperature parameter $\tau$ is introduced and we have:

$$q_k(x,c;\tau) = \frac{p(x \,|\, c; \theta_k)^{\frac{1}{T}}}{Z_{x,c}(\tau)}, \qquad Z_{x,c} = \sum_{s=1}^K p(x \,|\, c; \theta_s)^{\frac{1}{\tau}}, \tag{33}$$

where the temperature $t \mapsto \tau(t)$ follows a decreasing schedule, typically $\tau(t) = \tau(0)\rho^t$ with $\rho < 1$ and $\tau(0) > 0$ where $t$ in the training step. At high temperatures, training is distributed more evenly across all hypotheses, which helps to prevent collapse. As $\tau \to 0$, the method converges to the greedy WTA setup, which can maximize (oracle) performance provided that all hypotheses have been sufficiently trained. In the Audio Captioning experiments, Annealed MCL was trained with $\tau(0) = 1.0$, $\rho = 0.999$, and we switched back to vanilla WTA when the temperature reached $10^{-6}$.

### E.4 DECODING METHODS

Formally, Maximum-A-Posteriori decoding in the context of language modeling consist, given (fixed) parameters $\theta$ of finding sequences $\hat{x} = (\hat{x}_1, \ldots, \hat{x}_T) \in \mathcal{V}^T$ that maximizes $p_\theta(\hat{x} \,|\, c) = p_\theta(\hat{x}_1 \,|\, c) \prod_{t=2}^T p_\theta(\hat{x}_t \,|\, \hat{x}_{<t}, c)$ given a context $c \in \mathcal{C}$. Because an exhaustive search of the most likely sequence given $c$ and $\theta$ would be intractable, we used the following heuristics.

**Greedy & Beam Search.** Given a beam size (or beam width) $B$, beam search [46] proceeds as follows. First, compute the $B$ most likely tokens $\hat{x}_1^1, \ldots, \hat{x}_1^B$ from the distribution $p_\theta(\hat{x}_1 \,|\, c)$. Next, run a forward pass for each candidate $\hat{x}_1^i$ through the language model to obtain $B$ distributions $p_\theta(\cdot \,|\, \hat{x}_1^k, c)$ (for $k = 1, \ldots, B$), each over the vocabulary $\mathcal{V}$, yielding $B \times |\mathcal{V}|$ candidate probabilities. From these $B \times |\mathcal{V}|$ values, select the top-$B$ tokens to form the next candidates $\hat{x}_2^1, \ldots, \hat{x}_2^B$, while keeping track of the preceding tokens in each beam. Repeat this procedure for $t = 1, \ldots, T$ to produce $B$ sequences $\hat{x}^1, \ldots, \hat{x}^B \in \mathcal{V}^T$. Greedy search is the special case $B = 1$, i.e., at each step only the top-1 token is chosen: $\hat{x}_{t+1}^1 = \operatorname{argmax}_{x_{t+1}} p_\theta(x_{t+1} \,|\, \hat{x}_{<t})$. While this procedure is quite effective and reliable in practice, Beam Search is known to yield low diversity, returning candidates that differ only slightly near the ends of their decoding paths when asked for multiple outputs [84].

**Diverse Beam Search.** Diverse Beam Search [84] is an alternative to Beam Search in which the beam set of size $B$ is partitioned into $G$ disjoint groups of equal size $B' = B/G$. The goal is to maximize the likelihood within each group while encouraging dissimilarity across groups. Let the set for group $g$ at time $t$ be $X_t^g \triangleq \{\hat{x}_{1:t}^{b+(g-1)B'} \,|\, b = 1, \ldots, B'\}$. At each step $t$, and for each group

$g = 1, \ldots, G$ in ascending order, $X_t^g$ is updated via

$$X_t^g = \arg\max_{x_{1:t}^1, \ldots, x_{1:t}^{B'}} \sum_{b=1}^{B'} \left[ \log p_\theta(x_{1:t}^b \,|\, c) + \lambda \sum_{h=1}^{g-1} \Delta(x_{1:t}^b, X_t^h) \right], \qquad (34)$$

thereby trading off sequence likelihood and dissimilarity to the previously finalized groups, with diversity penalty $\lambda > 0$. In practice, the dissimilarity decomposes as $\Delta(\hat{x}_{1:t}, X_t^h) = \sum_{u_{1:t} \in X_t^h} \delta(\hat{x}_{1:t}, u_{1:t})$, where $\delta$ is a pairwise dissimilarity. In the Transformers implementation, $\delta$ uses a Hamming penalty at the current step, $\delta(\hat{x}_{1:t}, u_{1:t}) = \mathbf{1}[\hat{x}_t \neq u_t]$, so tokens already chosen at position $t$ by earlier groups are penalized.

**Test-time Augmentation.** TTA involves applying to the context $c$ a random perturbation $\mathcal{T}_\phi : \mathcal{C} \to \mathcal{C}$, parameterized by $\phi$ that controls the augmentation strength. We generate $K$ perturbed contexts $\tilde{c}_1, \ldots, \tilde{c}_K \sim \mathcal{T}_\phi(c)$ and perform MAP generation via Beam Search with beam size $\frac{B}{K}$ for each perturbed context. In our audio captioning experiments, we implement TTA using SpecAugment [65], where $\phi$ consists of two parameters: the percentage of time and frequency bands masked in the input spectrogram.

**Remarks on the decoding setup for each training method.** Beam Search was used as the decoding strategy for all three training methods described in Section E.3. To ensure a fair comparison under a fixed computational budget (i.e., a comparable number of forward passes), we adjust the beam size depending on the method:

- LoRA-MLE with BS or DBS uses a beam of size $B$, where $B \geq K$ if $K$ sequences are to be returned. For comparability, LoRA-MLE with TTA uses a beam size of $B/K$.

- LoRA-MCL decodes each of the $K$ hypotheses with a beam of size $B/K$, and each hypothesis returns a single sequence.

- LoRA-MoE was evaluated with two decoding strategies:

    1. *MLE Decoding*, where LoRA-MoE is treated as a single-hypothesis model with LoRA-MoE layers integrated into the base architecture. In this case, MAP decoding with beam size $B$ yields up to $K$ sequences (as in LoRA-MLE).
    2. *Stochastic Router Decoding*, following Zuo et al. [107], where the expert index is sampled randomly from the router's mixing-weight distribution at each layer. Each forward pass uses a beam size of $B/K$, returning one sequence.

Each of the above approaches is presented in the main submission results, except for Stochastic Router, which was omitted for conciseness due to its poor performance. For completeness, its results are provided in Tables 5, 7, and 9 of the Appendix.

### E.5 Parallelization over the hypotheses in LoRA-MCL

A naive implementation of MCL for winner selection, as in Section 3.2, may require a loop over the $K$ hypotheses in the batch to determine the winner associated with each index of the batch, which would drastically slow down training.

To alleviate this issue, we propose the following methodology. Let $\mathbf{x} \in \mathbb{R}^{b \times \mathcal{T} \times d}$ denote the input of the transformer architecture, where $b$ is the batch size, $\mathcal{T} \triangleq T + \tau$ is the total sequence length, and $d$ is the number of features. We duplicate $\mathbf{x}$, $K$ times along the batch size dimension to get $\mathbf{x}' = (\mathbf{x}_1, \ldots, \mathbf{x}_K) \in \mathbb{R}^{bK \times \mathcal{T} \times d}$.

LoRA units at layer $\ell$ are then computed as:

$$\begin{bmatrix} \mathbf{x}_1 \\ \mathbf{x}_2 \\ \vdots \\ \mathbf{x}_K \end{bmatrix} \leftarrow \begin{bmatrix} B_\ell^1 A_\ell^1 & 0 & 0 & 0 \\ 0 & B_\ell^2 A_\ell^2 & 0 & 0 \\ \vdots & \vdots & \ddots & \vdots \\ 0 & 0 & 0 & B_\ell^K A_\ell^K \end{bmatrix} \begin{bmatrix} \mathbf{x}_1 \\ \mathbf{x}_2 \\ \vdots \\ \mathbf{x}_K \end{bmatrix} + \begin{bmatrix} f_\theta^\ell(\mathbf{x}_1) \\ f_\theta^\ell(\mathbf{x}_2) \\ \vdots \\ f_\theta^\ell(\mathbf{x}_K) \end{bmatrix}, \qquad (35)$$

where $f_\theta^\ell$ is the base model (whose parameters remain frozen during training). In practice, this computation is conducted using a Group Convolution operation.[2] While this duplication virtually multiplies the batch size by $K$, the memory overhead remains manageable, assuming that $r \ll d$. The next section is dedicated to the inner properties of the proposed approach.

### E.6 AUDIO CAPTIONING EXPERIMENTS

#### E.6.1 EXPERIMENTAL SETUP

**Architecture and training.** We used the Instructed version of Qwen-2-Audio [6] as the base model, which features $\sim 8.4$ billion parameters. We trained using `bfloat16` precision. We used LoRA adapters applied to the $Q, K, V$ linear projections of the attention modules, and the upside and downside projections of the feedforward blocks, for all the transformer blocks, which include both the audio encoder and language model decoder. We used a rank $r$, with $r = \alpha = 8$ unless otherwise stated, with dropout equal to $0.1$ (enabled during training, and disabled during inference). We trained with a batch size of 2, with AdamW optimizer [45] (with $\beta_1 = 0.9$, and $\beta_2 = 0.98$), weight decay of $0.05$, using a cosine scheduler with minimum learning rate of $10^{-6}$ and maximum learning rate of $10^{-5}$, with a warmup ratio of $0.1$. Gradient clipping is used with a maximum gradient norm of $1.0$. The validation loss was computed once every epoch.

#### E.6.2 ADDITIONAL RESULTS

**Evaluating with Sampling-based decoding.** We also report results using sampling-based decoding methods in Tables 6 and 8 for Clotho and AudioCaps. Specifically, we use Top-$k$ sampling with $k = 50$, Top-$p$ sampling with $p = 0.95$, and Typical sampling with a threshold of $0.95$. For all sampling methods, we apply a repetition penalty of $1.1$, following [33]. In these experiments, the temperature was set to $\eta = 1.0$.

We observe that both `LoRA-MLE` and `LoRA-MCL` yield significantly higher diversity than MAP decoding, albeit at the cost of reduced output quality. This trade-off is evident in Tables 5 and 7. On AudioCaps, `LoRA-MCL` shows a slight improvement in both quality and diversity compared to `LoRA-MLE`. On Clotho, it provides a small gain in diversity, while quality slightly favors `LoRA-MLE`. These findings highlight the need for further evaluation with different annealing schedules to better characterize the quality–diversity trade-off in sampling-based decoding. Moreover, a deeper study of how the number of generated hypotheses influences sampling quality and its implications for test-time inference scaling with `LoRA-MCL` [102] is left for future work.

---

[2]This can be done by first reshaping $\mathbf{x}'$ to shape $(b \times \mathcal{T} \times Kd)$ and applying a `nn.Conv1d(`$Kd$`, `$Kd$`, kernel_size=1, groups=`$K$`)` following the PyTorch layer implementation, reshaping back $\mathbf{x}'$ to shape $(Kb \times \mathcal{T} \times d)$ before adding the base model output, and then repeating at the next LoRA unit.

Table 5: **Results for Clotho with 5 hypotheses and MAP Decoding.** 'BS', 'DBS', 'SR' and 'TTA' stand for beam search, diverse beam search, stochastic router, and test-time augmentation respectively.

| Training | Decoding | Beam | $Div_2$ | $mBLEU_4$ | $BLEU_1$ | $BLEU_4$ | METEOR | $ROUGE_L$ | sBERT | $CIDEr_D$ | SPICE | SPIDEr |
|---|---|---|---|---|---|---|---|---|---|---|---|---|
| LoRA-MLE ($r=8$) | BS | 5 | 0.365 | 0.822 | 0.656 | 0.137 | 0.228 | 0.445 | 0.575 | 0.626 | 0.174 | 0.394 |
| LoRA-MLE ($r=40$) | BS | 5 | 0.367 | 0.818 | 0.643 | 0.115 | 0.226 | 0.435 | 0.570 | 0.595 | 0.172 | 0.376 |
| LoRA-MLE ($r=8$) | BS | 10 | 0.391 | 0.783 | 0.661 | 0.142 | 0.236 | 0.453 | 0.578 | 0.646 | 0.181 | 0.406 |
| LoRA-MLE ($r=40$) | BS | 10 | 0.397 | 0.777 | 0.647 | 0.127 | 0.232 | 0.445 | 0.573 | 0.615 | 0.177 | 0.388 |
| LoRA-MLE ($r=8$) | BS | 25 | 0.405 | 0.759 | 0.658 | 0.144 | 0.236 | 0.455 | 0.579 | 0.648 | 0.182 | 0.407 |
| LoRA-MLE ($r=40$) | BS | 25 | 0.415 | 0.746 | 0.648 | 0.131 | 0.234 | 0.447 | 0.578 | 0.625 | 0.181 | 0.395 |
| LoRA-MLE ($r=8$) | DBS ($\lambda=0.8$) | 5 | 0.605 | 0.446 | 0.686 | 0.147 | 0.241 | 0.471 | 0.601 | 0.678 | 0.193 | 0.423 |
| LoRA-MLE ($r=40$) | DBS ($\lambda=0.8$) | 5 | 0.613 | 0.440 | 0.677 | 0.140 | 0.239 | 0.463 | 0.599 | 0.659 | 0.194 | 0.414 |
| LoRA-MLE ($r=8$) | DBS ($\lambda=1.0$) | 5 | 0.625 | 0.417 | 0.685 | 0.148 | 0.242 | 0.470 | **0.602** | 0.681 | 0.194 | 0.425 |
| LoRA-MLE ($r=40$) | DBS ($\lambda=1.0$) | 5 | 0.634 | 0.407 | 0.678 | 0.142 | 0.239 | 0.463 | 0.600 | 0.660 | 0.193 | 0.414 |
| LoRA-MLE ($r=8$) | DBS ($\lambda=0.8$) | 10 | 0.487 | 0.712 | 0.670 | 0.143 | 0.238 | 0.460 | 0.594 | 0.656 | 0.191 | 0.414 |
| LoRA-MLE ($r=40$) | DBS ($\lambda=0.8$) | 10 | 0.499 | 0.694 | 0.661 | 0.132 | 0.235 | 0.448 | 0.590 | 0.634 | 0.187 | 0.401 |
| LoRA-MLE ($r=8$) | DBS ($\lambda=1.0$) | 10 | 0.491 | 0.708 | 0.671 | 0.143 | 0.238 | 0.456 | 0.595 | 0.662 | 0.192 | 0.417 |
| LoRA-MLE ($r=40$) | DBS ($\lambda=1.0$) | 10 | 0.501 | 0.696 | 0.659 | 0.131 | 0.234 | 0.444 | 0.591 | 0.629 | 0.188 | 0.397 |
| LoRA-MLE ($r=8$) | DBS ($\lambda=0.8$) | 25 | 0.368 | 0.818 | 0.653 | 0.135 | 0.228 | 0.444 | 0.575 | 0.626 | 0.176 | 0.394 |
| LoRA-MLE ($r=40$) | DBS ($\lambda=0.8$) | 25 | 0.374 | 0.811 | 0.644 | 0.115 | 0.226 | 0.435 | 0.571 | 0.594 | 0.174 | 0.377 |
| LoRA-MLE ($r=8$) | DBS ($\lambda=1.0$) | 25 | 0.368 | 0.818 | 0.652 | 0.134 | 0.228 | 0.443 | 0.574 | 0.622 | 0.174 | 0.392 |
| LoRA-MLE ($r=40$) | DBS ($\lambda=1.0$) | 25 | 0.372 | 0.812 | 0.643 | 0.115 | 0.226 | 0.435 | 0.571 | 0.592 | 0.174 | 0.376 |
| LoRA-MLE ($r=8$) | TTA BS ($\phi_1$) | 1 | 0.443 | 0.699 | 0.652 | 0.114 | 0.225 | 0.440 | 0.581 | 0.608 | 0.174 | 0.383 |
| LoRA-MLE ($r=8$) | TTA BS ($\phi_2$) | 1 | 0.540 | 0.558 | 0.663 | 0.130 | 0.230 | 0.453 | 0.588 | 0.634 | 0.179 | 0.397 |
| LoRA-MLE ($r=8$) | TTA BS ($\phi_3$) | 1 | **0.642** | **0.404** | 0.653 | 0.118 | 0.225 | 0.441 | 0.581 | 0.596 | 0.174 | 0.376 |
| LoRA-MLE ($r=8$) | TTA BS ($\phi_1$) | 5 | 0.412 | 0.745 | 0.655 | 0.137 | 0.232 | 0.452 | 0.580 | 0.637 | 0.181 | 0.402 |
| LoRA-MLE ($r=8$) | TTA BS ($\phi_2$) | 5 | 0.516 | 0.593 | 0.681 | 0.156 | 0.243 | 0.470 | 0.591 | 0.685 | 0.194 | 0.430 |
| LoRA-MLE ($r=8$) | TTA BS ($\phi_3$) | 5 | 0.619 | 0.445 | 0.675 | 0.148 | 0.236 | 0.463 | 0.586 | 0.648 | 0.186 | 0.407 |
| LoRA-MoE | BS | 5 | 0.363 | 0.823 | 0.650 | 0.131 | 0.228 | 0.443 | 0.571 | 0.614 | 0.173 | 0.387 |
| LoRA-MoE | BS | 10 | 0.395 | 0.783 | 0.659 | 0.137 | 0.236 | 0.452 | 0.579 | 0.643 | 0.181 | 0.405 |
| LoRA-MoE | BS | 25 | 0.408 | 0.757 | 0.657 | 0.140 | 0.237 | 0.454 | 0.579 | 0.643 | 0.184 | 0.405 |
| LoRA-MoE | DBS ($\lambda=0.8$) | 5 | 0.611 | 0.441 | 0.682 | 0.143 | 0.241 | 0.468 | 0.599 | 0.668 | 0.194 | 0.418 |
| LoRA-MoE | DBS ($\lambda=1.0$) | 5 | 0.630 | 0.410 | 0.682 | 0.147 | 0.241 | 0.467 | 0.600 | 0.675 | 0.195 | 0.422 |
| LoRA-MoE | DBS ($\lambda=0.8$) | 10 | 0.490 | 0.706 | 0.670 | 0.142 | 0.238 | 0.459 | 0.593 | 0.662 | 0.192 | 0.416 |
| LoRA-MoE | DBS ($\lambda=1.0$) | 10 | 0.494 | 0.705 | 0.668 | 0.138 | 0.237 | 0.456 | 0.592 | 0.650 | 0.191 | 0.410 |
| LoRA-MoE | DBS ($\lambda=0.8$) | 25 | 0.370 | 0.814 | 0.650 | 0.128 | 0.228 | 0.445 | 0.572 | 0.613 | 0.175 | 0.386 |
| LoRA-MoE | DBS ($\lambda=1.0$) | 25 | 0.370 | 0.814 | 0.651 | 0.129 | 0.229 | 0.445 | 0.573 | 0.620 | 0.174 | 0.390 |
| LoRA-MoE | SR BS | 1 | 0.308 | 0.878 | 0.604 | 0.090 | 0.203 | 0.406 | 0.554 | 0.515 | 0.153 | 0.330 |
| LoRA-MoE | SR BS | 2 | 0.317 | 0.868 | 0.629 | 0.116 | 0.217 | 0.427 | 0.564 | 0.576 | 0.163 | 0.364 |
| LoRA-MoE | SR BS | 5 | 0.309 | 0.880 | 0.620 | 0.109 | 0.217 | 0.424 | 0.562 | 0.561 | 0.162 | 0.357 |
| LoRA-MCL ($\varepsilon=0.0005$) | BS | 1 | 0.605 | 0.440 | 0.675 | 0.135 | 0.233 | 0.458 | 0.597 | 0.654 | 0.187 | 0.408 |
| LoRA-MCL ($\varepsilon=0.05$) | BS | 1 | 0.617 | 0.415 | 0.680 | 0.130 | 0.239 | 0.463 | 0.601 | 0.662 | 0.190 | 0.414 |
| LoRA-MCL (annealed) | BS | 1 | 0.621 | 0.415 | 0.673 | 0.131 | 0.237 | 0.458 | 0.599 | 0.665 | 0.189 | 0.415 |
| LoRA-MCL ($\varepsilon=0.0005$) | BS | 2 | 0.591 | 0.461 | 0.688 | 0.154 | 0.242 | 0.472 | 0.598 | 0.688 | 0.192 | 0.428 |
| LoRA-MCL ($\varepsilon=0.05$) | BS | 2 | 0.593 | 0.452 | **0.692** | 0.160 | 0.247 | **0.481** | 0.599 | 0.694 | 0.193 | 0.431 |
| LoRA-MCL (annealed) | BS | 2 | 0.595 | 0.456 | 0.687 | 0.158 | 0.245 | 0.472 | 0.599 | 0.698 | 0.196 | 0.435 |
| LoRA-MCL ($\varepsilon=0.0005$) | BS | 5 | 0.581 | 0.488 | 0.687 | 0.156 | 0.244 | 0.476 | 0.599 | 0.700 | 0.196 | 0.436 |
| LoRA-MCL ($\varepsilon=0.05$) | BS | 5 | 0.581 | 0.478 | 0.689 | 0.162 | **0.249** | 0.480 | 0.599 | 0.697 | 0.196 | 0.434 |
| LoRA-MCL (annealed) | BS | 5 | 0.584 | 0.478 | 0.689 | **0.168** | 0.246 | 0.477 | 0.600 | **0.711** | **0.199** | **0.443** |

Table 6: **Results for Clotho with 5 hypotheses and Sampling-based Decoding.**

| Training | Decoding | $Div_2$ | $mBLEU_4$ | $BLEU_4$ | METEOR | $ROUGE_L$ | sBERT | $CIDEr_D$ | SPICE | SPIDEr |
|---|---|---|---|---|---|---|---|---|---|---|
| LoRA-MLE ($r=8$) | Top-$k$ sampling | 0.813 | 0.109 | 0.066 | **0.215** | 0.402 | 0.593 | 0.497 | **0.176** | 0.323 |
| LoRA-MLE ($r=8$) | Typical $p$ sampling | 0.812 | 0.111 | 0.073 | 0.212 | 0.403 | 0.586 | 0.516 | 0.173 | **0.331** |
| LoRA-MLE ($r=8$) | Nucleus (Top-$p$) sampling | 0.812 | 0.112 | 0.073 | 0.212 | 0.403 | 0.586 | 0.516 | 0.173 | 0.330 |
| LoRA-MCL (annealed) | Top-$k$ sampling | 0.819 | **0.097** | 0.074 | 0.212 | 0.403 | **0.597** | 0.507 | 0.175 | 0.327 |
| LoRA-MCL (annealed) | Typical $p$ sampling | 0.819 | 0.103 | 0.072 | 0.211 | 0.402 | 0.590 | 0.509 | 0.172 | 0.327 |
| LoRA-MCL (annealed) | Nucleus (Top-$p$) sampling | **0.820** | 0.098 | **0.074** | 0.212 | 0.403 | 0.589 | 0.507 | 0.171 | 0.325 |

### E.6.3 QUALITATIVE EXAMPLES

We provide some qualitative examples of the predictions on AudioCaps. Here, LoRA-MCL uses $\varepsilon = 0.05$, $B = 5$ and $K = 5$.

*Example 1.* References:

- A large truck driving by as an emergency siren wails and truck horn honks
- A wailing siren fades, a motor sputters, then the siren resumes accompanied by blaring horns
- An emergency siren ringing with car horn honking
- A fire truck engine runs and the siren is blowing but stops, traffic is present, the fire truck horn honks twice, and the siren begins again
- A fire engine with a siren fading then another loud siren

**LoRA-MCL.**
{A large truck driving by as an emergency siren wails and truck horn honks }

Table 7: **Results for AudioCaps with 5 hypotheses and MAP Decoding.**

| Training | Decoding | Beam | $\text{Div}_2$ | $\text{mBLEU}_4$ | $\text{BLEU}_4$ | METEOR | $\text{ROUGE}_L$ | sBERT | $\text{CIDEr}_D$ | SPICE | SPIDEr |
|---|---|---|---|---|---|---|---|---|---|---|---|
| LoRA-MLE ($r = 8$) | BS | 5 | 0.395 | 0.764 | 0.267 | 0.377 | 0.606 | 0.700 | 1.121 | 0.250 | 0.668 |
| LoRA-MLE ($r = 40$) | BS | 5 | 0.392 | 0.773 | 0.280 | 0.382 | 0.606 | 0.701 | 1.144 | 0.251 | 0.681 |
| LoRA-MLE ($r = 8$) | BS | 10 | 0.410 | 0.746 | 0.286 | 0.385 | 0.610 | 0.704 | 1.157 | 0.256 | 0.689 |
| LoRA-MLE ($r = 40$) | BS | 10 | 0.407 | 0.747 | 0.298 | 0.382 | 0.610 | 0.704 | 1.151 | 0.255 | 0.686 |
| LoRA-MLE ($r = 8$) | BS | 25 | 0.417 | 0.732 | 0.281 | 0.383 | 0.609 | 0.706 | 1.144 | 0.258 | 0.683 |
| LoRA-MLE ($r = 40$) | BS | 25 | 0.415 | 0.735 | 0.289 | 0.384 | 0.611 | 0.706 | 1.152 | 0.260 | 0.688 |
| LoRA-MLE ($r = 8$) | DBS ($\lambda = 0.8$) | 5 | 0.553 | 0.448 | 0.268 | 0.378 | 0.610 | 0.708 | 1.117 | 0.248 | 0.662 |
| LoRA-MLE ($r = 40$) | DBS ($\lambda = 0.8$) | 5 | 0.557 | 0.444 | 0.263 | 0.380 | 0.612 | 0.707 | 1.130 | 0.248 | 0.669 |
| LoRA-MLE ($r = 8$) | DBS ($\lambda = 1.0$) | 5 | 0.580 | 0.403 | 0.275 | 0.376 | 0.612 | 0.708 | 1.128 | 0.249 | 0.667 |
| LoRA-MLE ($r = 40$) | DBS ($\lambda = 1.0$) | 5 | 0.580 | 0.403 | 0.268 | 0.378 | 0.614 | 0.709 | 1.138 | 0.250 | 0.672 |
| LoRA-MLE ($r = 8$) | DBS ($\lambda = 0.8$) | 10 | 0.481 | 0.684 | 0.274 | 0.373 | 0.606 | 0.709 | 1.114 | 0.259 | 0.665 |
| LoRA-MLE ($r = 40$) | DBS ($\lambda = 0.8$) | 10 | 0.481 | 0.683 | 0.278 | 0.380 | 0.610 | 0.710 | 1.124 | 0.257 | 0.670 |
| LoRA-MLE ($r = 8$) | DBS ($\lambda = 1.0$) | 10 | 0.486 | 0.678 | 0.276 | 0.372 | 0.607 | 0.710 | 1.115 | 0.259 | 0.665 |
| LoRA-MLE ($r = 40$) | DBS ($\lambda = 1.0$) | 10 | 0.492 | 0.675 | 0.276 | 0.376 | 0.609 | 0.709 | 1.118 | 0.256 | 0.666 |
| LoRA-MLE ($r = 8$) | DBS ($\lambda = 0.8$) | 25 | 0.402 | 0.756 | 0.273 | 0.376 | 0.607 | 0.703 | 1.129 | 0.251 | 0.673 |
| LoRA-MLE ($r = 40$) | DBS ($\lambda = 0.8$) | 25 | 0.399 | 0.763 | 0.282 | 0.381 | 0.605 | 0.702 | 1.139 | 0.253 | 0.680 |
| LoRA-MLE ($r = 8$) | DBS ($\lambda = 1.0$) | 25 | 0.402 | 0.757 | 0.271 | 0.377 | 0.608 | 0.703 | 1.128 | 0.251 | 0.672 |
| LoRA-MLE ($r = 40$) | DBS ($\lambda = 1.0$) | 25 | 0.398 | 0.765 | 0.282 | 0.381 | 0.606 | 0.701 | 1.136 | 0.252 | 0.678 |
| LoRA-MLE ($r = 8$) | TTA BS ($\phi_1$) | 1 | 0.385 | 0.731 | 0.198 | 0.343 | 0.574 | 0.693 | 0.969 | 0.226 | 0.584 |
| LoRA-MLE ($r = 8$) | TTA BS ($\phi_2$) | 1 | 0.479 | 0.588 | 0.207 | 0.352 | 0.586 | 0.695 | 0.992 | 0.231 | 0.597 |
| LoRA-MLE ($r = 8$) | TTA BS ($\phi_3$) | 1 | 0.576 | 0.438 | 0.185 | 0.332 | 0.567 | 0.683 | 0.916 | 0.218 | 0.550 |
| LoRA-MLE ($r = 8$) | TTA BS ($\phi_1$) | 5 | 0.395 | 0.733 | 0.241 | 0.358 | 0.592 | 0.700 | 1.057 | 0.242 | 0.636 |
| LoRA-MLE ($r = 8$) | TTA BS ($\phi_2$) | 5 | 0.491 | 0.590 | 0.260 | 0.360 | 0.597 | 0.703 | 1.047 | 0.246 | 0.630 |
| LoRA-MLE ($r = 8$) | TTA BS ($\phi_3$) | 5 | **0.596** | 0.435 | 0.251 | 0.347 | 0.586 | 0.693 | 1.005 | 0.235 | 0.605 |
| LoRA-MoE | BS | 5 | 0.396 | 0.766 | 0.274 | 0.381 | 0.608 | 0.702 | 1.129 | 0.252 | 0.674 |
| LoRA-MoE | BS | 10 | 0.411 | 0.746 | 0.288 | 0.385 | 0.613 | 0.703 | 1.161 | 0.256 | 0.692 |
| LoRA-MoE | BS | 25 | 0.415 | 0.736 | 0.287 | 0.385 | 0.612 | 0.705 | 1.154 | 0.258 | 0.689 |
| LoRA-MoE | DBS ($\lambda = 0.8$) | 5 | 0.555 | 0.443 | 0.275 | 0.379 | 0.611 | 0.707 | 1.136 | 0.248 | 0.670 |
| LoRA-MoE | DBS ($\lambda = 1.0$) | 5 | 0.578 | 0.409 | 0.268 | 0.377 | 0.609 | 0.708 | 1.130 | 0.247 | 0.667 |
| LoRA-MoE | DBS ($\lambda = 0.8$) | 10 | 0.484 | 0.677 | 0.279 | 0.374 | 0.608 | 0.709 | 1.122 | 0.256 | 0.667 |
| LoRA-MoE | DBS ($\lambda = 1.0$) | 10 | 0.488 | 0.675 | 0.283 | 0.374 | 0.608 | 0.710 | 1.119 | 0.257 | 0.666 |
| LoRA-MoE | DBS ($\lambda = 0.8$) | 25 | 0.402 | 0.757 | 0.275 | 0.382 | 0.609 | 0.701 | 1.135 | 0.253 | 0.676 |
| LoRA-MoE | DBS ($\lambda = 1.0$) | 25 | 0.401 | 0.759 | 0.275 | 0.382 | 0.609 | 0.701 | 1.133 | 0.253 | 0.675 |
| LoRA-MoE | SR BS | 1 | 0.256 | 0.909 | 0.165 | 0.318 | 0.538 | 0.670 | 0.865 | 0.200 | 0.526 |
| LoRA-MoE | SR BS | 2 | 0.269 | 0.895 | 0.195 | 0.326 | 0.549 | 0.675 | 0.905 | 0.210 | 0.551 |
| LoRA-MoE | SR BS | 5 | 0.261 | 0.907 | 0.196 | 0.331 | 0.550 | 0.678 | 0.925 | 0.216 | 0.563 |
| LoRA-MCL ($\varepsilon = 0.0005$) | BS | 1 | 0.580 | **0.374** | 0.259 | 0.377 | 0.613 | 0.711 | 1.119 | 0.248 | 0.662 |
| LoRA-MCL ($\varepsilon = 0.05$) | BS | 1 | 0.551 | 0.427 | 0.273 | 0.382 | 0.622 | 0.712 | 1.181 | 0.253 | 0.695 |
| LoRA-MCL (annealed) | BS | 1 | 0.570 | 0.397 | 0.275 | 0.379 | 0.615 | 0.713 | 1.149 | 0.255 | 0.680 |
| LoRA-MCL ($\varepsilon = 0.0005$) | BS | 2 | 0.575 | 0.401 | 0.277 | 0.385 | 0.626 | 0.712 | 1.152 | 0.258 | 0.684 |
| LoRA-MCL ($\varepsilon = 0.05$) | BS | 2 | 0.544 | 0.464 | 0.298 | 0.394 | 0.631 | 0.712 | 1.209 | 0.260 | 0.714 |
| LoRA-MCL (annealed) | BS | 2 | 0.574 | 0.411 | 0.309 | 0.392 | 0.632 | 0.714 | 1.205 | 0.264 | 0.712 |
| LoRA-MCL ($\varepsilon = 0.0005$) | BS | 5 | 0.579 | 0.410 | 0.306 | 0.392 | 0.631 | 0.712 | 1.190 | 0.263 | 0.706 |
| LoRA-MCL ($\varepsilon = 0.05$) | BS | 5 | 0.542 | 0.491 | 0.309 | **0.399** | 0.636 | **0.715** | **1.237** | 0.265 | **0.728** |
| LoRA-MCL (annealed) | BS | 5 | 0.575 | 0.423 | **0.315** | 0.398 | **0.636** | 0.714 | 1.211 | **0.268** | 0.716 |

Table 8: **Results for AudioCaps with 5 hypotheses and Sampling-based Decoding.**

| Training | Decoding | $\text{Div}_2$ | $\text{mBLEU}_4$ | $\text{BLEU}_4$ | METEOR | $\text{ROUGE}_L$ | sBERT | $\text{CIDEr}_D$ | SPICE | SPIDEr |
|---|---|---|---|---|---|---|---|---|---|---|
| LoRA-MLE ($r = 8$) | Top-$k$ sampling | 0.711 | 0.173 | 0.169 | 0.335 | 0.549 | 0.701 | 0.906 | 0.229 | 0.543 |
| LoRA-MLE ($r = 8$) | Nucleus (Top-$p$) sampling | 0.697 | 0.193 | 0.180 | **0.344** | 0.564 | 0.704 | 0.940 | 0.234 | 0.563 |
| LoRA-MLE ($r = 8$) | Typical $p$ sampling | 0.699 | 0.189 | 0.182 | 0.343 | 0.564 | 0.704 | 0.945 | 0.233 | 0.566 |
| LoRA-MCL (annealed) | Top-$k$ sampling | **0.735** | **0.149** | 0.175 | 0.342 | 0.568 | **0.705** | 0.938 | 0.233 | 0.563 |
| LoRA-MCL (annealed) | Nucleus (Top-$p$) sampling | 0.724 | 0.163 | 0.182 | 0.340 | 0.566 | 0.702 | 0.950 | 0.235 | 0.569 |
| LoRA-MCL (annealed) | Typical $p$ sampling | 0.724 | 0.160 | **0.186** | 0.341 | **0.568** | 0.703 | **0.953** | 0.235 | **0.570** |

{A wailing siren fades, a motor sputters, then the siren resumes accompanied by blaring horns}
{An emergency siren ringing with car horn honking}
{A fire truck engine runs and the siren is blowing but stops, traffic is present, the fire truck horn honks twice, and the siren begins again}
{A fire engine with a siren fading then another loud siren}.

*Example 2.* References:

- A man speaks as birds chirp and dogs bark

- A man is speaking as birds are squawking, and a dog barks

- A man talks while several animals make noises in the background

- A man speaking followed by dogs barking alongside chimps screaming and birds chirping

- A man speaking as monkeys scream and dogs bark followed by birds cawing in the distance

**LoRA-MCL.**
{A man speaking and dogs barking}
{A man is narrating and a dog is barking in the background}
{A man is speaking and dogs are barking}
{Man speaking with dog barking in the background}
{A man speaks with dogs barking and birds chirping in the background}

*Example 3.* References:

- An engine running and wind with various speech in the background

- A motorboat engine operating as a crowd of people talk followed by metal creaking and a man speaking

- A large motor is running smoothly, water is splashing, people are talking in the background, and an adult male speaks in the distance

- A ship engine running as a crowd of people talk followed by a ship hull creaking as wind blows heavily into a microphone

- Outdoor noise from a water vehicle as people are talking

**LoRA-MCL.**
{A boat motor is running and people are talking in the background}
{An aircraft engine running as people talk in the background}
{An engine is running and people are talking}
{An engine running consistently with people talking in the background}
{Humming of an engine with distant murmuring}

### E.7 IMAGE CAPTIONING EXPERIMENTS

### E.7.1 EXPERIMENTAL SETUP

We used LLaVA 1.6 with Vicuna-7B [103] for the LLM, as the base model, which features 7.1 billion parameters. We used the official codebase for the implementation. We trained using `bfloat16` precision. We used LoRA adapters applied to the $Q, K, V$, up and down linear projections of each block of the language model. We used a rank $r$, with $r = 8$ and $\alpha = 32$ unless otherwise stated, with dropout equal to $0.1$ (enabled during training, and disabled during inference). We trained each model with a batch size of 8, with AdamW optimizer [45] (with $\beta_1 = 0.9$, and $\beta_2 = 0.98$), weight decay of $0.05$, using a cosine scheduler with maximum learning rate of $10^{-4}$, with a warmup ratio of $0.1$. Gradient clipping is used with a maximum gradient norm of $1.0$. We used 1 epoch for training, where we duplicated the image as many times as the number of its captions, such that the model sees exactly one time each caption.

Table 9: **Quality and Diversity Evaluation on TextCaps with** 3 **candidates**. For each of the presented metrics, higher is better (↑) except for mBLEU-4 (↓). LoRA-MCL is trained with $\varepsilon = 0.1$, $r = 8$ and $\alpha = 32$. LoRA-MLE is trained with $r = 24$ and $\alpha = 96$. For completeness, we also trained LoRA-MLE with $r = 8$ and $\alpha = 32$ in the rows marked with $^\dagger$.

| Training | Decoding | Beam | Div$_2$ | mBLEU$_4$ | BLEU$_1$ | BLEU$_4$ | METEOR | ROUGE$_L$ | sBERT | CIDEr$_D$ | SPICE | SPIDEr |
|---|---|---|---|---|---|---|---|---|---|---|---|---|
| LoRA-MLE | BS | 3 | 0.509 | 0.688 | 0.802 | 0.318 | 0.315 | 0.580 | 0.670 | 1.517 | 0.244 | 0.873 |
| LoRA-MLE | BS | 6 | 0.457 | 0.786 | 0.795 | 0.338 | 0.326 | 0.583 | 0.671 | 1.557 | 0.246 | 0.895 |
| LoRA-MLE$^\dagger$ | BS | 3 | 0.421 | 0.833 | 0.788 | 0.315 | 0.317 | 0.573 | 0.670 | 1.517 | 0.241 | 0.874 |
| LoRA-MLE$^\dagger$ | BS | 6 | 0.456 | 0.784 | 0.796 | 0.339 | 0.327 | 0.585 | 0.672 | 1.572 | 0.248 | 0.903 |
| LoRA-MLE | DBS ($\lambda = 0.5$) | 3 | 0.600 | 0.529 | 0.818 | 0.345 | 0.325 | 0.596 | 0.684 | 1.571 | 0.253 | 0.902 |
| LoRA-MLE | DBS ($\lambda = 0.8$) | 3 | 0.655 | 0.437 | 0.824 | 0.349 | 0.327 | 0.601 | 0.686 | 1.590 | 0.251 | 0.909 |
| LoRA-MLE | DBS ($\lambda = 1.0$) | 3 | **0.669** | **0.416** | 0.822 | 0.348 | 0.326 | 0.599 | 0.685 | 1.586 | 0.250 | 0.906 |
| LoRA-MLE | DBS ($\lambda = 0.5$) | 6 | 0.532 | 0.694 | 0.813 | 0.344 | 0.328 | 0.595 | 0.681 | 1.580 | 0.253 | 0.908 |
| LoRA-MLE | DBS ($\lambda = 0.8$) | 6 | 0.549 | 0.671 | 0.812 | 0.341 | 0.328 | 0.593 | 0.681 | 1.573 | 0.251 | 0.903 |
| LoRA-MLE | DBS ($\lambda = 1.0$) | 6 | 0.553 | 0.666 | 0.812 | 0.340 | 0.328 | 0.592 | 0.680 | 1.577 | 0.250 | 0.904 |
| LoRA-MLE$^\dagger$ | DBS ($\lambda = 0.5$) | 3 | 0.597 | 0.531 | 0.821 | 0.346 | 0.326 | 0.596 | 0.685 | 1.589 | 0.255 | 0.912 |
| LoRA-MLE$^\dagger$ | DBS ($\lambda = 0.8$) | 3 | 0.597 | 0.531 | 0.821 | 0.346 | 0.326 | 0.596 | 0.685 | 1.589 | 0.255 | 0.912 |
| LoRA-MLE$^\dagger$ | DBS ($\lambda = 1.0$) | 3 | 0.665 | 0.425 | 0.827 | 0.357 | 0.327 | 0.601 | 0.686 | 1.601 | 0.252 | 0.915 |
| LoRA-MLE$^\dagger$ | DBS ($\lambda = 0.5$) | 6 | 0.528 | 0.697 | 0.808 | 0.343 | 0.329 | 0.594 | 0.681 | 1.586 | 0.253 | 0.911 |
| LoRA-MLE$^\dagger$ | DBS ($\lambda = 0.8$) | 6 | 0.542 | 0.681 | 0.810 | 0.340 | 0.329 | 0.593 | 0.680 | 1.583 | 0.253 | 0.909 |
| LoRA-MLE$^\dagger$ | DBS ($\lambda = 1.0$) | 6 | 0.551 | 0.671 | 0.810 | 0.341 | 0.328 | 0.592 | 0.680 | 1.584 | 0.251 | 0.908 |
| LoRA-MoE | DBS ($\lambda = 0.5$) | 3 | 0.600 | 0.529 | 0.822 | 0.347 | 0.327 | 0.598 | 0.684 | 1.610 | **0.258** | 0.923 |
| LoRA-MoE | DBS ($\lambda = 0.8$) | 3 | 0.654 | 0.441 | **0.828** | 0.353 | 0.327 | 0.603 | 0.685 | 1.616 | 0.254 | 0.924 |
| LoRA-MoE | DBS ($\lambda = 1.0$) | 3 | 0.666 | 0.421 | **0.828** | 0.354 | 0.328 | 0.602 | 0.685 | 1.622 | 0.253 | 0.926 |
| LoRA-MoE | DBS ($\lambda = 0.5$) | 6 | 0.530 | 0.698 | 0.814 | 0.348 | 0.331 | 0.595 | 0.681 | 1.607 | 0.257 | 0.923 |
| LoRA-MoE | DBS ($\lambda = 0.8$) | 6 | 0.545 | 0.678 | 0.813 | 0.349 | 0.330 | 0.596 | 0.680 | 1.608 | 0.255 | 0.922 |
| LoRA-MoE | DBS ($\lambda = 1.0$) | 6 | 0.551 | 0.670 | 0.813 | 0.346 | 0.330 | 0.596 | 0.679 | 1.607 | 0.254 | 0.921 |
| LoRA-MoE | SR BS | 1 | 0.556 | 0.597 | 0.809 | 0.315 | 0.311 | 0.576 | 0.678 | 1.541 | 0.243 | 0.883 |
| LoRA-MCL | BS | 1 | 0.599 | 0.520 | **0.828** | 0.344 | 0.330 | 0.597 | **0.690** | **1.674** | 0.255 | **0.955** |
| LoRA-MCL | BS | 2 | 0.618 | 0.490 | 0.824 | **0.360** | **0.333** | **0.604** | 0.687 | 1.627 | **0.258** | 0.932 |

### E.7.2 ADDITIONAL RESULTS

Table 9 presents the results of LoRA-MCL and LoRA-MLE. Almost all the metrics show the same trend in which LoRA-MCL outperforms LoRA-MLE with Beam Search (BS) and Diverse Beam Search (DBS) in terms of quality, although DBS produces more varied outputs. Depending on the rank ($r = 8$ or $r = 24$), we found that setting $\lambda = 1.0$ or $\lambda = 0.8$, respectively, yields the best quality scores for DBS. Similar to our experiments on Audio Captioning, increasing the rank of LoRA-MLE results in a slight degradation in performance but improves diversity. Additionally, increasing the number of beams in BS with LoRA-MLE results in slightly improved performance but reduced diversity. In contrast, increasing the number of beams in DBS with LoRA-MLE (with fixed $\lambda$) leads to declines in both quality and diversity. Interestingly, with LoRA-MCL, increasing the number of beams enhances both performance and diversity here.

### E.7.3 ARTIFICIAL MULTILINGUAL DATASET CREATION

To evaluate the behaviour of LoRA-MCL under a multi-modal distribution, we simulated an artificial bi-modal dataset by automatically translating half of the captions from English to French using T5-small [69], while keeping the prompts in English. More specifically, we randomly sampled half of the images and translated their five associated captions. All the training parameters are the same as those in the experiments on the original TextCaps dataset, except the learning rate, which we set at $2 \times 10^{-5}$ (as the maximum value in the scheduler) in both the LoRA-MLE and LoRA-MCL.

During evaluation, to assess which head is considered as the winner (for the head specialization analysis), we selected the one that maximizes the SPIDEr score over the references of the given sample.

### E.7.4 QUALITATIVE EXAMPLES

In this section, we show some qualitative examples (image–predicted caption pairs) that highlight the behavior of LoRA-MCL compared to a baseline model (LoRA-MLE with $r = 24$). When the scene contains numerous objects or pieces of information, LoRA-MCL (through its different heads) covers a wider variety of descriptions than diverse beam search decoding. We performed inference with LoRA-MCL using greedy decoding, while we used diverse beam search ($\lambda = 0.8$ and using 3 beams) for the baseline model.

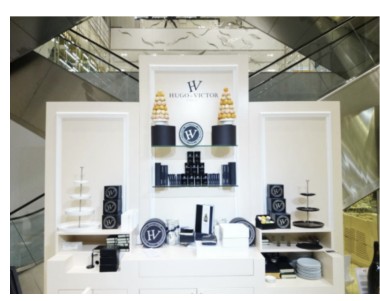

**Baseline.**
{A display of Hugo Victor products on a table.}
{A display for Hug Victor with a sign that says Hug Victor.}
{a store display for Hugo Victor with many items

**LoRA-MCL.**
{A display of Hugo Victor products is set up in a store.}
{A store display for Hugo Victor with a shelf full of items.}
{A store display for Hugo Victor features a variety of plates, bowls, and books.}

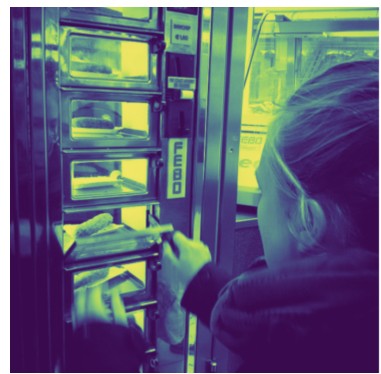

**Baseline.**
{a person getting food from a vending machine with the number 680 on it}
{A woman is getting food from a vending machine that says "Fresh" on it.}
{A girl puts a sandwich into a vending machine that says "Fresh" on it.}

**LoRA-MCL.**
{A girl is putting food into a vending machine that says FEEDO.}
{A woman is putting money into a vending machine that says 100.}
{A girl is getting a hot dog from a vending machine.}

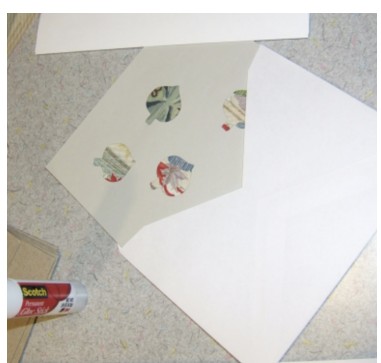

**Baseline.**
{ A Scotch brand glue stick is on a table.}
{A can of Scotch branded glue is on a table.}

{a scotch tape that is on the ground}

**LoRA-MCL.**
{A bottle of Scotch tape sits on a table.}
{A bottle of Scotch tape sits on a table next to a piece of paper.}
{A Scotch tape is on the table next to a piece of paper.}

### E.8 DIVERSE MACHINE TRANSLATION

We fine-tuned ALMA-7B (based on LLaMA-2-7B [81]) for two epochs on the human-written data collected by Xu et al. [93], which includes the WMT'17–WMT'20 test sets and the development and test sets from Flores-200 [7]. Our fine-tuning setup follows Xu et al. [93], except that we employ bf16 precision instead of fp16. We use $r = \frac{\alpha}{2} = 16$, a warm-up ratio of $0.01$, a maximum sequence length of $512$, an effective batch size of $256$, and a peak learning rate of $2 \times 10^{-5}$ with an inverse square-root scheduler.

We follow the quality–diversity evaluation protocol of Shen et al. [76], computing BLEU scores with the SacreBLEU library. The reported metrics are Leave-One-Out BLEU (Loo-BLEU) and Pairwise-BLEU (see Section 4 of [76]), using the official codebase. Leave-One-Out BLEU (denoted simply as BLEU in [76]) measures the overall quality of the hypothesis set by computing the corpus-level BLEU for each hypothesis against the reference set (higher is better). Pairwise-BLEU considers only the predicted hypotheses, computing BLEU scores for each pair of candidates (lower indicates greater diversity).

### E.9 COMPUTATION DETAILS

We run the experiments mostly on H100 NVIDIA GPUs with $80$ GB of RAM. Trainings and inferences were launched on a single GPU for the Audio Captioning experiments, and up to $8$ GPUs

for the Image Captioning experiments. The total computing resources used for this project, including failed experiments, amount to approximately 20,000 GPU hours.

### E.10 USE OF LARGE LANGUAGE MODELS

We used LLMs assistants in the preparation of this work. They helped to polish the writing (e.g., improving clarity, grammar, and style without altering the content) and serving as a coding assistant (e.g., visualization, debugging).

