# OpenReview forum: "Multiple Choice Learning of Low Rank Adapters for Language Modeling"
_ICLR.cc/2026/Conference — Submitted to ICLR 2026_

### Official Review · Reviewer_MhHA · 2025-10-15

**Soundness:** 4
**Presentation:** 3
**Contribution:** 3
**Rating:** 6
**Confidence:** 3

**Summary:**

The paper introduces LoRA-MCL, a training scheme for language models that combines Multiple Choice Learning (MCL) with Low-Rank Adapters (LoRA) with the goal of generating diverse and plausible sentence continuations. ​ Traditional language modeling often struggles with capturing the inherent ambiguity of multi-modal data distributions. ​ LoRA-MCL addresses this problem by using multiple hypotheses and a Winner-Takes-All (WTA) loss to specialize each hypothesis in different modes of the data distribution. The method is computationally efficient by leveraging parallelization techniques. ​ Experiments on audio captioning, image captioning, and machine translation demonstrate that LoRA-MCL achieves a good balance between diversity and quality, outperforming traditional methods in challenging scenarios. ​

**Strengths:**

1. The combination of Multiple Choice Learning (MCL) with Low-Rank Adaptation (LoRA) seems to be novel.

2. The authors provide a solid theoretical analysis proving propositions and demonstrating how LoRA-MCL captures the modes of a mixture distribution and linking it to the Winner-Takes-All (WTA) loss and hard-EM optimization. ​

3. LoRA-MCL is evaluated across diverse tasks, including audio captioning, image captioning, and machine translation.

4. The method achieves a good balance between diversity and quality in generated outputs compared to baselines LoRA-MLE and LoRA-MoE, as demonstrated through experiments. ​

5. The use of LoRA adapters and parallelization techniques ensures that the method is computationally efficient, making it scalable for large language models. ​

**Weaknesses:**

1. While this is a nice theoretical paper with apparently solid proofs, I feel the efficacy may be limited because the experiments mainly focus on LoRA-X without considering the large body of work in audio captioning, image captioning, and machine translation.

2. LoRA-MCL is sensitive to hyperparameters like the relaxation parameter (ε) and temperature scheduler (τ). The paper acknowledges that. ​

3. The proposed method involves multiple hypotheses and parallelization techniques, which may increase implementation complexity compared to simpler approaches like LoRA-MLE. This issue may be alleviated with the release of working codes.

4. While the paper evaluates sampling-based decoding methods, the results show mixed performance, and further exploration of annealing schedules and their impact on quality-diversity trade-offs is needed. ​

5. Despite efforts to mitigate collapse, the paper admits that some hypotheses may still be under-trained, which could limit the diversity of outputs. ​

6. The experiments focus on comparing LoRA-MCL, LoRA-MLE in audio captioning, image captioning, and machine translation. However  the comparison does not include or address other SOTA in these tasks.

**Questions:**

1. How robust is LoRA-MCL to variations in the relaxation parameter (ε) and temperature scheduler (τ)? Are there guidelines for selecting these parameters for new tasks?
2. Have you considered integrating advanced LoRA variants (e.g., LoRA+ or MixLoRA) into LoRA-MCL? If so, what challenges or benefits do you anticipate?
3. Sampling-Based Decoding: The results for sampling-based decoding methods are mixed. Could you elaborate on how LoRA-MCL could be optimized for these methods, particularly in terms of annealing schedules? ​
4. How well do you expect LoRA-MCL to perform on other tasks like dialogue generation, summarization, or code generation?
5. Despite the use of relaxed WTA and annealed MCL, is there a risk of model collapse in scenarios with highly imbalanced modes? ​ How could this be further mitigated? ​

**Details Of Ethics Concerns:**

No concerns.

---

> ### Author Response · Authors · 2025-11-22
>
> We thank the reviewer for their remarks, which allow to improve the quality of the paper.
>
> > How robust is LoRA-MCL to variations in the relaxation parameter ($ε$) and temperature scheduler ($τ$)? Are there guidelines for selecting these parameters for new tasks?
>
> We provide an ablation on the relaxation parameter ε and the decay rate $ρ$ of the temperature scheduler ($\tau(t) = ρ^{t} \tau_0$) in Table E, which are analyzed below.
>
> **Effect of $ε$.** Let $\ell_k = - \log p(x \mid c, \theta_{k})$ denote the NLL of hypothesis $k$ for a pair $(c,x)$. Recall that the relaxed MCL loss for such a pair is
> $$ (1-ε)  \ell_{k}^{\\star} + \frac{ε}{K-1} \sum_{k=1, k \neq k^{\\star}}^{K} \ell_k = (1 -  \frac{K ε}{K-1}) \ell_{k}^{\\star} + \frac{ε}{K-1} \sum_{k=1}^{K} \ell_k,$$
> where $k^{\\star}$ is the winner hypothesis (See Eq.4).
>
> We see that the first term is a force that pushes the hypothesis $k$ toward the winner hypothesis, while the second term assigns equal weight to each hypothesis, pulling them toward the barycenter of the conditional distribution (See Figure 1 in [66]). The first term vanishes when $ε = \frac{K-1}{K}$, which provides an upper bound on the value of ε. In practice, when choosing ε for a new task, the smaller is the better unless a collapse is observed; we recommand trying $\varepsilon \in [0,0.1]$. We run LoRA-MCL (relaxed) with $ε \in \\{0.0005, 0.05, 0.1, 0.3, 0.5, 0.8\\}$ on AudioCaps (AC) and Clotho (CL) in Table A with $K = 5$ (LoRA-MCL "relaxed" rows). We observe that outside the "small" ε regime (0.0005, 0.05), increasing ε significantly degrades the diversity (e.g., mBLEU₄ from 0.410 with $ε$ = 0.0005 on AudioCaps to 0.963 with $ε$ = 0.8). For quality (as measured by Oracle SPIDEr), performance also tends to degrade outside the small-ε regime. This occurs because large $\varepsilon$ diminishes the benefits of MCL training, causing the model to behave increasingly like LoRA-MLE, where the heads eventually become uniformised.
>
> **Effect of the temperature scheduler.** LoRA-MCL annealed is based on equations 4 and 5. The annealed method relies on the theoretical foundation of deterministic annealing (e.g., [A, 66]), which predicts "phase transition" phenomena in which the hypotheses "split" at specific temperature levels, namely critical temperatures, to explore hierarchically different modes of the conditional distribution. Given a fixed number of training steps, the temperature schedule needs to be neither too slow, otherwise the final training step may still occur above the critical temperature, nor too fast, in which cause the model does not fully benefit from annealing and collapse may still occur. We provide an ablation on the decay rate $\rho$ in Table A (see the annealed rows) for $ρ \in \\{0.9, 0.995, 0.999, 0.9999\\}$. Performance remains good accross most of this range, except for $\rho = 0.9999$ where we observe a degradation of diversity and perforance. We attribute this to the final temperature remaining above the critical temperature.
>
> *Table A: Ablation on the relaxation parameters in LoRA-MCL Evaluation on AudioCaps (AC) and Clotho (CL).*
> |Training|Decoding|Beam|mBLEU₄ (AC) ($\downarrow$)|mBLEU₄ (CL) ($\downarrow$)|SPIDEr (AC) ($\uparrow$)|SPIDEr (CL) ($\uparrow$)|
> |-|-|-|-|-|-|-|
> |LoRA-MLE ($r$=8)|DBS (λ=0.8)|5|0.448|*0.446*|0.662|0.423|
> |LoRA-MCL (relaxed, $ε$=0.0005)|BS|5|**0.410**|0.488|0.706|*0.436*|
> |LoRA-MCL (relaxed, $ε$=0.05)|BS|5|0.491|0.478|0.728|0.434|
> |LoRA-MCL (relaxed, $ε$=0.1)|BS|5|0.502|0.526|**0.738**|0.435|
> |LoRA-MCL (relaxed, $ε$=0.3)|BS|5|0.624|0.643|0.700|0.421|
> |LoRA-MCL (relaxed, $ε$=0.5)|BS|5|0.766|0.790|0.634|0.390|
> |LoRA-MCL (relaxed, $ε$=0.8)|BS|5|0.963|0.952|0.508|0.329|
> |LoRA-MCL (annealed, $ρ$=0.9)|BS|5|0.458|**0.432**|0.701|0.429|
> |LoRA-MCL (annealed, $ρ$=0.995)|BS|5|*0.422*|0.490|*0.729*|*0.438*|
> |LoRA-MCL (annealed, $ρ$=0.999)|BS|5|0.423|0.478|0.716|**0.443**|
> |LoRA-MCL (annealed, $ρ$=0.9999)|BS|5|0.830|0.947|0.610|0.331|

---

> ### Author Response · Authors · 2025-11-22
>
> > Have you considered integrating advanced LoRA variants into LoRA-MCL? If so, what challenges or benefits do you anticipate?
>
> Yes. As noted in the limitations, we believe advances in the LoRA literature, such as LoRA+ [21], which uses a larger learning rate for $B$ compared to $A$, could be integrated into LoRA-MCL in a fairly straightforward manner. Other strategies, such as LoRA-One [101], which proposes deterministic initialization of $A$ and $B$ (based on the SVD of the loss gradient), may also be interesting to explore. However, this latter approach may require additional adaptation, since diversity in the initializations of the $(A_k)_k$ modules is a key ingredient for LoRA-MCL work. We plan to explore these directions as further work.
>
> > The results for sampling-based decoding methods are mixed. Could you elaborate on how LoRA-MCL could be optimized for these methods, particularly in terms of annealing schedules? ​
>
> We believe that improving the sampling-based decoding results is possible with a better temperature scheduler, which may benefit MAP decoding results as well. There are some desired properties for the LoRA-MCL scheduler, such as its convergence towards 0 while being slow enough. In the context of stochastic simulated annealing [B], Hajek [C] came up with a (slow-enough) condition on the temperature scheduler to reach a global minimum of the training objective. In practice, using the slowest
> possible scheduler won't necessarily work for medium-sized datasets because finding a good optimum of the training objective doesn't always imply good generalisation. This requires further investigation.
>
> > How well do you expect LoRA-MCL to perform on other tasks like dialogue generation, summarization, or code generation?
>
> We believe LoRA-MCL is well-suited for open-ended text generation tasks, including those discussed in the paper, as well as dialogue generation, summarization, and code generation. However, LoRA-MCL may be less appropriate for task with fully deterministic targets and minimal ambiguity, such as multiple-choice question answering, where diversity across hypotheses offers limited benefit.
>
> > Despite the use of relaxed WTA and annealed MCL, is there a risk of model collapse in scenarios with highly imbalanced modes? How could this be further mitigated?
>
> In principle, for scenarios involving "rare" modes, we expect one (or several) hypotheses to specialize in these regions to capture them (See Figure B.2 in [39]). We believe the risk of model collapse is already mitigated with relaxed WTA and annealed MCL, provided that the relaxation parameters are properly chosen. As discussed earlier, we provide guidance on selecting these parameters, though predicting optimal values without experiments remains an open problem.

---

> ### Author Response · Authors · 2025-11-22
>
> > I feel the efficacy may be limited because the experiments mainly focus on LoRA-X without considering the large body of work in audio captioning, image captioning, and machine translation.
>
> Our goal in that paper is to adapt (multimodal) large language models with MCL in the context of LoRA to obtain a good compromise between quality and diversity in the context of text generation. We made comparison in fair settings where each baseline has the same architecture and training details. The models we considered (LLaVA 1.6, Qwen-2 Audio, ALMA) are among the strongest open MLLMs currently available. We agree that further comparisons may be insightful, but comparing with baselines with completely different architecture is hard, because it's tricky to tell if the difference in performance comes from the architecture itself or the training method. That said, we conducted additional experiments during the rebuttal period on both audio and image captioning to further contextualize our results. We provide the results below.
>
> **Audio Captioning.** We compare against [D] a diffusion-based method using retrieval-guided Langevin dynamics (DAC-RLD), which we identify as the strongest published method for diverse audio captioning on Clotho and AudioCaps with publicly available code and checkpoints. A VAE-based method [E] is also open-sourced (Clotho only) but underperforms DAC-RLD in both quality and diversity. Adversarial training has also been explored [56, 95], though these methods are outperformed by [D], and do not release code. We evaluate the pretrained DAC-RLD checkpoints using our setup: 5 candidate captions per audio (instead of 50 in the original work), oracle sentence-level quality metrics, and mBLEU₄ for diversity. For DAC-RLD, we run both with Beam Search and Nucleus sampling, as in the original work. Results (Table B) show that DAC-RLD achieves diversity comparable to LoRA-MCL with nucleus sampling (slightly better or worse depending on the dataset) but at the cost of substantially lower SPIDEr scores than LoRA-MCL on both Clotho and AudioCaps.
>
> *Table B: Comparison against SOTA in Diverse Audio Captioning. Oracle evaluation with K = 5 on AudioCaps (AC) and Clotho (CL) at 16kHz.*
> |Training|Decoding|Beam|mBLEU₄(AC)|mBLEU₄(CL)|SPIDEr(AC)|SPIDEr(CL)|
> |-|-|-|-|-|-|-|
> |DAC-RLD|Nucleus(p=0.95)|1|**0.157**|*0.150*|0.435|0.244|
> |DAC-RLD|BS|5|0.239|0.215|0.505|0.287|
> |LoRA-MCL(annealed)|Nucleus(p=0.95)|1|*0.163*|**0.098**|0.569|0.325|
> |LoRA-MCL(ε=0.05)|BS|5|0.491|0.478|**0.728**|*0.434*|
> |LoRA-MCL(annealed)|BS|5|0.423|0.478|*0.716*|**0.443**|
>
> **Image Captioning.** Several work explored improving caption diversity in image captioning [85, 86, 48]. Among those evaluating on TextCaps, [G] and [H] report results, but their performance is far below that of recent VLMs, such as PaLiGemma-3B [F]. For example, the reported best corpus CIDEr on TextCaps val are: [G] 76.6, [H] 95.5, PaLiGemma-3B (224x224): 127.48. We therefore compared against PaLiGemma-3B, which provides a fine-tuned version on Hugging Face. We evaluate it with DBS returning K = 3 hypotheses (Table C). PaliGemma is a strong baseline, arguably the state-of-the-art open-weight model on TextCaps. Note, however, that the comparison is not entirely fair, as PaLiGemma undergoes full-weight fine-tuning (instead of LoRA). Despite this, our method matches and even slightly improves its performance (SPIDEr 0.955 against 0.949). We also expect that applying LoRA-MCL to PaLiGemma could further improve the results.
>
> *Table C: Quality and Diversity Evaluation on TextCaps (K=3). Table caption is as in Table 2.*
> |Training|Decoding|Beam|mBLEU-4|SPIDEr|
> |-|-|-|-|-|
> |PaliGemma-3B(ft)|DBS(λ=0.8)|3|**0.467**|0.949|
> |LoRA-MCL(relaxed,ε=0.1)|BS|1|0.520|**0.955**|
>
> **References**
>
> [A] Rose, K., et al. Vector quantization by deterministic annealing. IEEE Transactions on Information theory, 2022.
>
> [B] Kirkpatrick, S. et al. Optimization by simulated annealing. Science, 1983
>
> [C] Hajek. Cooling schedules for optimal annealing. Mathematics of Operations Research, 1988
>
> [D] Zhu, Y. et al. "Diffusion-based diverse audio captioning with retrieval-guided Langevin dynamics." Information Fusion, 2025.
>
> [E] Zhang, Y. et al. "Generating Accurate and Diverse Audio Captions Through Variational Autoencoder Framework." IEEE SPL, 2024.
>
> [F] Beyer, L. et al. "Paligemma: A versatile 3b vlm for transfer." arXiv:2407.07726, 2024.
>
> [G] Zhang, W. et al. "Magic: Multimodal relational graph adversarial inference for diverse and unpaired text-based image captioning." AAAI, 2022.
>
> [H] Xu, G. et al. "Towards accurate text-based image captioning with content diversity exploration." CVPR, 2021.

---

> ### Comment · Reviewer_MhHA · 2025-11-24
> **Rebuttal OK. I would have raised my score from 6 to 7...**
>
> Thank you authors for your hard work on the detailed rebuttal. I would've raised my score from 6 to 7, but this year reviewers can only enter even-numbered scores. While I am still hesitant to give the paper an 8, I would like to see how other reviewers chime in, especially the negative review with high confidence after reading all other reviews and the authors rebuttal.

---

### Official Review · Reviewer_i36m · 2025-10-27

**Soundness:** 3
**Presentation:** 3
**Contribution:** 3
**Rating:** 2
**Confidence:** 4

**Summary:**

This paper introduces LoRA-MCL: applies Multiple Choice Learning with a Winner-Takes-All loss to K parallel LoRA adapters, enabling single-pass, diverse hypotheses while avoiding collapse. Across audio/image captioning and machine translation, achieves stronger quality-diversity trade-offs.

**Strengths:**

1. The motivation is clear and principled. By applying MCL with a Winner-Takes-All loss, the paper offers a practical recipe for training LLMs with multiple specialized hypotheses, leading to better learning dynamics and stronger outputs.
2. The LoRA-based design is both practical and efficient.
3. Results cover several multimodal tasks, spanning visual and audio captioning as well as machine translation.

**Weaknesses:**

1. The novelty is limited. The proposed method adds little beyond reformatting existing LoRA techniques into a multiple-choice scheme.
2. The evaluation centers on LoRA-MLE, making it difficult to assess superiority over stronger alternatives. Please include broader baselines (SOTA LLMs and multi-modal models) as well as MoE and multi-head variants.
3. The current diversity evaluation (limited to Div-n and mBLEU-n) is insufficient, as both are length-sensitive and mostly lexical. Please add semantic and distributional diversity metrics and report the quality-diversity trade-off. For quality, also include LLM-as-a-judge and a human-preference study.
4. Given that the proposed approach targets LLMs, the evaluation would be much stronger with a broader set of text-only benchmarks to demonstrate generality beyond the current settings.
5. The rationale for preferring LoRA over multi-headed outputs is insufficiently evidenced. The comparison centers on parameter count (=640M), while other claimed disadvantages are asserted without empirical backing. Please provide evidence beyond params.

**Questions:**

Please see the concerns detailed in the Weaknesses.

---

> ### Author Response · Authors · 2025-11-22
>
> We thank the reviewer for their detailed remarks. We provide here a detailed answer to the raised concerns.
>
> > The evaluation centers on LoRA-MLE, making it difficult to assess superiority over stronger alternatives. Please include broader baselines (SOTA LLMs and multi-modal models)
>
> First please note that appart from the different decoding methods for LoRA-MLE (BS, DBS, TTA, Sampling), we already report LoRA-MoE (Mixture of Experts) as a baseline (See Figure 2 and Table 2).
> We provide additional comparison with state-of-the-art in audio and image captioning:
>
> **Audio Captioning.** We compare against [B] a diffusion-based method using retrieval-guided Langevin dynamics (DAC-RLD), which we identify as the strongest published method for diverse audio captioning on Clotho and AudioCaps with publicly available code and checkpoints. A VAE-based method [D] is also open-sourced (Clotho only) but underperforms DAC-RLD in both quality and diversity. Adversarial training has also been explored [56, 95], though these methods are outperformed by [B], and do not release code. We evaluate the pretrained DAC-RLD checkpoints using our setup: 5 candidate captions per audio (instead of 50 in the original work), oracle sentence-level quality metrics, and mBLEU₄ for diversity. For DAC-RLD, we run both with Beam Search and Nucleus sampling, as in the original work. Results (Table A) show that DAC-RLD achieves diversity comparable to LoRA-MCL with nucleus sampling (slightly better or worse depending on the dataset) but at the cost of substantially lower SPIDEr scores than LoRA-MCL on both Clotho and AudioCaps.
>
> *Table A: Comparison against SOTA in Diverse Audio Captioning. Oracle evaluation with $K$ = 5 on AudioCaps (AC) and Clotho (CL) at 16kHz.*
> |Training|Decoding|Beam|mBLEU₄(AC)|mBLEU₄(CL)|SPIDEr(AC)|SPIDEr(CL)|
> |-|-|-|-|-|-|-|
> |DAC-RLD|Nucleus(p=0.95)|1|**0.157**|*0.150*|0.435|0.244|
> |DAC-RLD|BS|5|0.239|0.215|0.505|0.287|
> |LoRA-MCL(annealed)|Nucleus(p=0.95)|1|*0.163*|**0.098**|0.569|0.325|
> |LoRA-MCL(ε=0.05)|BS|5|0.491|0.478|**0.728**|*0.434*|
> |LoRA-MCL(annealed)|BS|5|0.423|0.478|*0.716*|**0.443**|
>
> **Image Captioning.** Several works explored improving caption diversity in image captioning [85, 86, 48]. Among those evaluating on TextCaps, [E] and [F] report results, but their performance is far below that of recent VLMs, such as PaLiGemma-3B [B]. For example, the reported best corpus CIDEr on TextCaps val are: [E] 76.6, [F] 95.5, PaLiGemma-3B (224x224): 127.48. We therefore compared against PaLiGemma-3B, which provides a fine-tuned version on Hugging Face. We evaluate it with DBS returning $K$ = 3 hypotheses (Table B). PaliGemma is a strong baseline, arguably the state-of-the-art open-weight model on TextCaps. Note, however, that the comparison is not entirely fair, as PaLiGemma undergoes full-weight fine-tuning (instead of LoRA). Despite this, our method matches and even slightly improves its performance (SPIDEr 0.955 against 0.949). We also expect that applying LoRA-MCL to PaLiGemma could further improve the results.
>
> *Table B: Quality and Diversity Evaluation on TextCaps (K=3). Table caption is as in Table 2.*
> |Training|Decoding|Beam|mBLEU-4|SPIDEr|
> |-|-|-|-|-|
> |PaliGemma-3B(ft)|DBS(λ=0.8)|3|**0.467**|0.949|
> |LoRA-MCL(relaxed,ε=0.1)|BS|1|0.520|**0.955**|

---

> ### Author Response · Authors · 2025-11-22
>
> > The current diversity evaluation (limited to Div-n and mBLEU-n) is insufficient, as both are length-sensitive and mostly lexical. Please add semantic and distributional diversity metrics and report the quality-diversity trade-off. For quality, also include LLM-as-a-judge and a human-preference study.
>
> As an additional semantic diversity metric, we used Self-Cider [85] (higher for more diverse). As a LLM-as-a-judge metric, we used ClairA [A] (using Phi-4-Mini 3.8B [G] as the Judge LLM) in audio captioning. We computed ClairA in oracle mode with $K = 5$ predictions, consistent with our other quality evaluations.
>
> Results on AudioCaps are shown in Table C. We observe a strong correlation between mBLEU₄ (↓) and Self-CIDEr (↑); substituting one for the other does not change the conclusions. Likewise, ClairA (↑) closely tracks SPIDEr (↑). We plan to integrate these new metrics into the next revision of the paper.
>
> *Table C: Results for AudioCaps with 5 hypotheses and MAP Decoding.*
> |Training|Decoding|Beam|mBLEU₄|Self-CIDEr|ClairA|SPIDEr|
> |-|-|-|-|-|-|-|
> |LoRA-MLE ($r$=8)|BS|5|0.764|0.639|0.642|0.668|
> |LoRA-MLE ($r$=8)|BS|10|0.746|0.649|0.652|0.689|
> |LoRA-MLE ($r$=8)|BS|25|0.732|0.657|0.656|0.683|
> |LoRA-MLE ($r$=8)|DBS (λ=0.8)|5|0.448|0.766|0.648|0.662|
> |LoRA-MLE ($r$=8)|DBS (λ=0.8)|10|0.684|0.713|0.663|0.665|
> |LoRA-MLE ($r$=8)|DBS (λ=0.8)|25|0.756|0.641|0.647|0.673|
> |LoRA-MoE|BS|5|0.766|0.638|0.644|0.674|
> |LoRA-MoE|BS|10|0.746|0.651|0.652|0.692|
> |LoRA-MoE|BS|25|0.736|0.656|0.659|0.689|
> |LoRA-MoE|DBS (λ=0.8)|5|0.443|0.769|0.650|0.670|
> |LoRA-MoE|DBS (λ=0.8)|10|0.677|0.715|0.661|0.667|
> |LoRA-MoE|DBS (λ=0.8)|25|0.757|0.643|0.649|0.676|
> |LoRA-MCL(ε=0.0005)|BS|1|**0.374**|**0.803**|0.648|0.662|
> |LoRA-MCL(ε=0.05)|BS|1|0.427|0.776|0.655|0.695|
> |LoRA-MCL(annealed)|BS|1|*0.397*|0.786|0.658|0.680|
> |LoRA-MCL(ε=0.0005)|BS|2|0.401|*0.800*|0.655|0.684|
> |LoRA-MCL(ε=0.05)|BS|2|0.464|0.762|0.665|0.714|
> |LoRA-MCL(annealed)|BS|2|0.411|0.782|0.667|0.712|
> |LoRA-MCL(ε=0.0005)|BS|5|0.410|0.799|0.659|0.706|
> |LoRA-MCL(ε=0.05)|BS|5|0.491|0.751|**0.673**|**0.728**|
> |LoRA-MCL(annealed)|BS|5|0.423|0.779|*0.672*|*0.716*|
>
> > Given that the proposed approach targets LLMs, the evaluation would be much stronger with a broader set of text-only benchmarks to demonstrate generality beyond the current settings.
>
> We thank the reviewer for the suggestion. Our work specifically targets ambiguous tasks in a LoRA-based setup. We already evaluated across synthetic data and on three modatilities (text, audio, vision) covering 6 datasets in total. We expect the method to extend naturally to broader free-form text generation tasks (e.g., summarization, open-ended generation), and we plan to explore these directions in future work.

---

> ### Author Response · Authors · 2025-11-22
>
> > The rationale for preferring LoRA over multi-headed outputs is insufficiently evidenced. The comparison centers on parameter count (=640M), while other claimed disadvantages are asserted without empirical backing. Please provide evidence beyond params.
>
> To address the request for evidence, we conducted additional comparisons with multi-head variants. In audio captioning, we duplicated the Language model head ("LMHead", ~640M params) $K$ times, froze the rest of the model and applied MCL training (without LoRA) with both relaxed and annealed variants. As an additional variant, showing that parameter count alone is not the only issue, we also duplicated an internal module (the multimodal projector "MMProj", ~5.2M parameters) and trained it with the same scheme. As discussed in the paper, multi-head initialization is not trivial. Copying the pretrained head yields no initialization diversity, while random reinitialization discards pretrained knowledge.
>
> Results are displayed in Table D. We find that
> (i) Random initialization (Init "Random") degrades quality for both MMProj and LMHead, and in the LMHead case, produces largely unintelligible captions with artificially high diversity.
> (ii) Copying the parameters of the pretrained model (Init "Copy") produces coherent outputs, but does not meaningfully benefit from MCL training due to amplified collapse risk, yielding significantly lower quality in practice.
> These findings show that multi-head approaches are not competitive with LoRA-MCL in a comparable training setup under LLM setting.
>
> *Table D: Comparison of LoRA-MCL with Multi-head fine-tuning on AudioCaps. Each method uses K = 5 and Beam Search decoding with Beam = 5*
> |Training|Init|mBLEU₄|SPIDEr|
> |-|-|-|-|
> |Multi-head (LMHead, relaxed)|Copy|0.489|0.561|
> |Multi-head (LMHead, annealed)|Copy|0.847|0.501|
> |Multi-head (LMHead, relaxed)|Random|**0.001**|0.002|
> |Multi-head (MMProj, relaxed)|Copy|0.530|0.394|
> |Multi-head (MMProj, relaxed)|Random|0.256|0.140|
> |LoRA-MCL (relaxed)|LoRA|0.491|**0.728**|
> |LoRA-MCL (annealed)|LoRA|0.423|0.716|
>
> **References**
>
> [A] Wu, T.-H. et al. "Clair-a: Leveraging large language models to judge audio captions." arXiv:2409.12962, 2024.
>
> [B] Zhu, Y. et al. "Diffusion-based diverse audio captioning with retrieval-guided Langevin dynamics." Information Fusion, 2025.
>
> [C] Beyer, L. et al. "Paligemma: A versatile 3b vlm for transfer." arXiv:2407.07726, 2024.
>
> [D] Zhang, Y. et al. "Generating Accurate and Diverse Audio Captions Through Variational Autoencoder Framework." IEEE SPL, 2024.
>
> [E] Zhang, W. et al. "Magic: Multimodal relational graph adversarial inference for diverse and unpaired text-based image captioning." AAAI, 2022.
>
> [F] Xu, G. et al. "Towards accurate text-based image captioning with content diversity exploration." CVPR, 2021.
>
> [G] Abouelenin, A. et al. Phi-4-mini technical report: Compact yet powerful multimodal language models via mixture-of-loras. arXiv:2503.01743, 2025

---

### Official Review · Reviewer_xYGJ · 2025-10-31

**Soundness:** 3
**Presentation:** 3
**Contribution:** 3
**Rating:** 8
**Confidence:** 3

**Summary:**

The authors propose a training scheme called LoRA-MCL for LLM-decoder based generation, where to introduce diversity in the generation, they run K LoRAs on the same base model to get diverse outputs. Their design is inspired by the insight that depending on context, the next word prediction can vary a lot such that different predictions are valid in different contexts. They propose that having multiple adapters can lead to each of them being able to predict the next word distribution with a different context, and hence ensure diversity. To train the system with K LoRAs, they propose a winner takes all (WTA) loss. They also demonstrate via experiments with synthetic sequences sampled from a mixture of markov chain distributions that MCL is able to model the underlying distribution more accurately than just using MLE loss using single LoRA. Finally, they compare the quality and diversity of generated outputs from LoRA based beamsearch and MCL based beamsearch, on real world tasks such as audio and image caption generation and machine translation. Results look strongest for audio captioning(Figure2) where MCL provides better diversity and quality. On image caption generation(Table2) and MT(Figure4), the results are a bit mixed where on image caption generation MCL gets higher quality but lesser diversity, and on MT it gets higher diversity but lesser quality.

**Strengths:**

- Excellent motivation and theoretical grounding of the problem by looking at next token prediction via the lens of modeling mixture of distributions.
- Proposes an efficient implementation by using custom architecture implementation of running K LoRAs in a single forward pass via batching and grouped convolution.
- Care taken to have fair comparison with baselines while keeping parity on inference budget by adjusting LoRA rank and beam widths accordingly (314-323)
- Quality vs. diversity trade-off is nice for audio captioning where we have better quality and diversity via MCL (Figure 2) The results are a bit mixed in other settings such as Table 2 where diversity decreases with respect to the baselines although the quality increases.

**Weaknesses:**

Despite the grouped convolution, you need to create a batch of the input sequence repeated K times for forward pass which increases FLOPs. While this would be faster than running K separate inferences in a for loop, it would still incur a cost with respect to throughput when deployed at scale where the GPU utilization is saturated.

**Questions:**

- In equation 6, $\mathcal{L}^{\text{WTA}}(\theta)$ has 2 formulas, is that for relaxed WTA and annealed WTA respectively?
- In Table 1, the maximum value of r in MLE is 40 but for MCL maximum value of K is 7 which should be equivalent to 7x8=56 ranked LoRA in MLE right? Should the value of K only go up to 5 for fair comparison here?
- In Table 2, it would help to add the value of K for each MCL row. Are we supposed to divide the number of beams present in MLE by MCL to infer the value of K? It is confusing to parse the results here.

---

> ### Author Response · Authors · 2025-11-22
>
> We are grateful to the reviewer for their positive feedback and insightful comments.
>
> > Despite the grouped convolution, you need to create a batch of the input sequence repeated K times for forward pass which increases FLOPs. While this would be faster than running K separate inferences in a for loop, it would still incur a cost with respect to throughput when deployed at scale where the GPU utilization is saturated.
>
> The reviewer is right; despite the optimized implementation described in Section 3.4, our method still increases training FLOPs compared to the baseline. However, inference FLOPs are identical with MLE with beam size $B$ when using a beam size $B/K$ for MCL.
>
> > $\mathcal{L}^{\mathrm{WTA}}(\theta)$ has 2 formulas, is that for relaxed WTA and annealed WTA respectively?
>
> Annealed and Relaxed WTA correspond both to equation 4 with different expressions for the $(q_k)_{k=1}^{K}$. For relaxed WTA, $q_k = 1 - ε$ for $k = k^{\\star}$ and $q_k = \frac{ε}{K-1}$ for $k ≠ k^{\\star}$. For annealed WTA, the expression of $q_k$ is given by equation 5. Equation 6 corresponds to the vanilla WTA loss when there is a perfect matching between hypotheses and modes, which is when the model is perfectly optimized. This will be made clearer in the next revision.
>
> > In Table 1, the maximum value of r in MLE is 40 but for MCL maximum value of K is 7 which should be equivalent to 7x8=56 ranked LoRA in MLE right? Should the value of K only go up to 5 for fair comparison here?
>
> The reviewer is indeed right.
> Following the reviewers' suggestion, we also include the results with $r \in \\{8×2, 8×3, 8×7\\}$ in Table A, to complete Table 1. These show that the trend on the evolution of the performance with respect to the rank is dataset-dependent; the performance of LoRA-MLE slightly improves and then stagnates on AudioCaps, and degrades on Clotho as $r$ increases.
>
> *Table A: Test Loss $(\downarrow)$ as a function of K.*
> | Training                         | $K$ | AudioCaps | Clotho |
> |----------------------------------|---|-----------|--------|
> | LoRA-MLE ($r$ = 8)               | 1 | 2.203     | 2.812  |
> | LoRA-MLE ($r$ = 8 × 2)           | 1 | 2.202     | 2.846  |
> | LoRA-MLE ($r$ = 8 × 3)           | 1 | 2.195     | 2.868  |
> | LoRA-MLE ($r$ = 8 × 5)           | 1 | 2.181     | 2.910  |
> | LoRA-MLE ($r$ = 8 × 7)           | 1 | 2.182     | 2.935  |
> | LoRA-MCL                       | 2 | 2.096     | 2.692  |
> | LoRA-MCL                       | 3 | 2.063     | 2.663  |
> | LoRA-MCL                       | 5 | 1.999     | 2.643  |
> | LoRA-MCL                     | 7 | **1.932** | **2.612** |
>
> > In Table 2, it would help to add the value of K for each MCL row. Are we supposed to divide the number of beams present in MLE by MCL to infer the value of K? It is confusing to parse the results here.
>
> In Table 2, the number of returned hypotheses is fixed to 3 for the whole table. For MCL rows, we therefore have $K = 3$ hypotheses, where a specified Beam size of $B$ means each hypothesis returns one sequence using beam size $B$.

---

### Official Review · Reviewer_i3JF · 2025-11-01

**Soundness:** 3
**Presentation:** 2
**Contribution:** 2
**Rating:** 6
**Confidence:** 3

**Summary:**

This paper proposes a method that integrates Multiple Choice Learning with LoRA to improve the ability of language models to generate diverse yet plausible text continuations. The key idea is to address the inherent multimodality of next-token prediction by training multiple LoRA adapters within the same model using a relaxed/annealed WTA loss. This design allows different adapters to specialize in different modes of the conditional output distribution while remaining parameter-efficient. The authors provide theoretical justification under a mixture-of-distributions assumption, validate the framework on mixtures of Markov chains. The authors conduct experiments on audio captioning, image captioning, and machine translation, showing improved quality–diversity trade-offs compared with LoRA-based baselines.

**Strengths:**

1. The paper adopts the Multiple Choice Learning framework with multiple LoRA modules to capture diverse yet plausible text continuations, and introduces a grouped parallel implementation that enables efficient training of all hypotheses with minimal memory overhead.

2. The paper provides theoretical analysis to demonstrate the effectiveness of the proposed loss function, and then verifies it using synthetic Mixture of Markov Chains data, showing that the approach can successfully recover distinct latent modes as expected.

3. The method is tested on various tasks, including image captioning, audio captioning, and machine translation, and shows promising results on certain tasks.

**Weaknesses:**

1. The method is relatively straightforward, employing Multiple Choice Learning with multiple LoRA modules and relying on existing mechanisms such as the WTA loss and its Relaxed/Annealed variants.
2. The proposed method shows limited improvement over other LoRA-based approaches in certain cases. For instance, as shown in Table 2, it performs notably worse than LoRA-MLE DBS (λ = 1.0) on the mBLEU-4 metric, which measures diversity, and does not demonstrate clear advantages on other evaluation metrics either.
3. Although the paper evaluates on multiple tasks such as image and audio captioning, it mainly compares with other LoRA-based baselines. Since the method aims to improve the trade-off between diversity and quality, comparisons with task-specific state-of-the-art methods would strengthen the empirical claims.

**Questions:**

1. During inference, does the model also adopt a WTA-like mechanism to select the response from the LoRA module with the highest probability, or do all K LoRA modules independently generate K separate outputs? Since each LoRA module is expected to specialize in a particular output type, could the authors provide evidence showing what each module has learned in the real tasks such as image or audio captioning?
2. Could the proposed method achieve better performance or improved efficiency compared with existing state-of-the-art methods that aim to enhance diversity in the tasks evaluated in this paper, such as image and audio captioning?
3. Regarding Weakness 2, I noticed that the paper mentions LoRA-MCL combined with DBS might alleviate this issue. I am curious whether there are concrete experiments showing the performance of LoRA-MCL when used with different decoding strategies, such as Beam Search or Diverse Beam Search.
4. In LoRA training, increasing the rank does not always improve performance and can sometimes cause overfitting or unstable optimization. Could the LoRA-MLE baseline be underperforming because the large-rank setup (rank =$K \times r$) leads to optimization instability, while other smaller ranks might perform better?

---

> ### Author Response · Authors · 2025-11-22
>
> We thank the reviewer for their detailed feedback. We provide here a detailed answer to the raised concerns.
>
> > During inference, [...] do all K LoRA modules independently generate K separate outputs?
>
> During inference, the $K$ LoRA modules independently generate $K$ output sequences. This is useful in ambiguous tasks where a single prediction is insufficient.
>
> > Since each LoRA module is expected to specialize [...], could the authors provide evidence showing what each module has learned in the real tasks [...]?
>
> In Section 5.3.2, we demonstrated specialization in image captioning using a two-head model trained on captions randomly assigned to French or English (each with probability 1/2). One of the hypotheses learned French and the other English, illustrating unsupervised specialization. Nevertheless, identifying each head's style from qualitative samples alone is difficult in the general case.
>
> Going further, to address the reviewer's question and quantify specialization, we embed every generated caption on the evaluation set (for hypotheses $k = 1,\dots,K$ and each example $i=1,\dots,N$) using a Sentence-BERT model (StyleDistance [A]). We trained a linear SVM on this space to predict which head produced each caption, using a 70/30 train-test split on captions.
>
> In the French/English experiment, the SVM achieves 100% test accuracy for the two-head LoRA-MCL model shown in Figure 3, confirming specialization. Outside this controlled setup, we repeated this analysis on captions from the LoRA-MCL trained on AudioCaps with $K=5$. The SVM reaches 63.7\% test-accuracy, over three times better than random choice ($20 \%$), indicating a clear specialization of the heads. In contrast, applying the same procedure to captions from the LoRA-MLE baseline (with DBS decoding, $\lambda = 0.8$, and beam size = 5) yielded 23.2\% accuracy on the test set after SVM fitting, close to random choice and showing no evidence of specialization, likewise LoRA-MoE (yielding 19.8\% accuracy in the same setup).

---

> > ### Author Response · Authors · 2025-11-22
> >
> > > the paper [...] mainly compares with other LoRA-based baselines. [...] Could the proposed method achieve better performance or improved efficiency compared with existing state-of-the-art methods that aim to enhance diversity [...] ?
> >
> > We provide additional comparison with state-of-the-art in audio and image captioning:
> >
> > **Audio Captioning.** We compare against [B] a diffusion-based method using retrieval-guided Langevin dynamics (DAC-RLD), which we identify as the strongest published method for diverse audio captioning on Clotho and AudioCaps with publicly available code and checkpoints. A VAE-based method [D] is also open-sourced (Clotho only) but underperforms DAC-RLD in both quality and diversity. Adversarial training has also been explored [56, 95], though these methods are outperformed by [B], and do not release code. We evaluate the pretrained DAC-RLD checkpoints using our setup: 5 candidate captions per audio (instead of 50 in the original work), oracle sentence-level quality metrics, and mBLEU₄ for diversity. For DAC-RLD, we run both with Beam Search and Nucleus sampling, as in the original work. Results (Table A) show that DAC-RLD achieves diversity comparable to LoRA-MCL with nucleus sampling (slightly better or worse depending on the dataset) but at the cost of substantially lower SPIDEr scores than LoRA-MCL on both Clotho and AudioCaps.
> >
> > *Table A: Comparison against SOTA in Diverse Audio Captioning. Oracle evaluation with K = 5 on AudioCaps (AC) and Clotho (CL) at 16kHz.*
> > |Training|Decoding|Beam|mBLEU₄(AC)|mBLEU₄(CL)|SPIDEr(AC)|SPIDEr(CL)|
> > |-|-|-|-|-|-|-|
> > |DAC-RLD|Nucleus(p=0.95)|1|**0.157**|*0.150*|0.435|0.244|
> > |DAC-RLD|BS|5|0.239|0.215|0.505|0.287|
> > |LoRA-MCL(annealed)|Nucleus(p=0.95)|1|*0.163*|**0.098**|0.569|0.325|
> > |LoRA-MCL(ε=0.05)|BS|5|0.491|0.478|**0.728**|*0.434*|
> > |LoRA-MCL(annealed)|BS|5|0.423|0.478|*0.716*|**0.443**|
> >
> > **Image Captioning.** Several works explored improving caption diversity in image captioning [85, 86, 48]. Among those evaluating on TextCaps, [E] and [F] report results, but their performance is far below that of recent VLMs, such as PaLiGemma-3B [B]. For example, the reported best corpus CIDEr on TextCaps val are: [E] 76.6, [F] 95.5, PaLiGemma-3B (224x224): 127.48. We therefore compared against PaLiGemma-3B, which provides a fine-tuned version on Hugging Face. We evaluate it with DBS returning K = 3 hypotheses (Table B). PaliGemma is a strong baseline, arguably the state-of-the-art open-weight model on TextCaps. Note, however, that the comparison is not entirely fair, as PaLiGemma undergoes full-weight fine-tuning (instead of LoRA). Despite this, our method matches and even slightly improves its performance (SPIDEr 0.955 against 0.949). We also expect that applying LoRA-MCL to PaLiGemma could further improve the results.
> >
> > *Table B: Quality and Diversity Evaluation on TextCaps (K=3). Table caption is as in Table 2.*
> > |Training|Decoding|Beam|mBLEU-4|SPIDEr|
> > |-|-|-|-|-|
> > |PaliGemma-3B(ft)|DBS(λ=0.8)|3|**0.467**|0.949|
> > |LoRA-MCL(relaxed,ε=0.1)|BS|1|0.520|**0.955**|

---

> > > ### Author Response · Authors · 2025-11-22
> > >
> > > > I noticed that the paper mentions LoRA-MCL combined with DBS might alleviate this issue. I am curious whether there are concrete experiments showing the performance of LoRA-MCL when used with different decoding strategies, such as Beam Search or Diverse Beam Search.
> > >
> > > In our experiments, we used Beam Search as the decoding method for LoRA-MCL. There are two possible ways to apply Diverse Beam Search (DBS) to LoRA-MCL:
> > > (i) applying DBS within each head, but this would be equivalent to standard Beam Search when each head returns only a single sequence; or
> > > (ii) enforcing dissimilarity across the sequences generated by the different MCL heads, inspired by [G].
> > > The second approach would be interesting to explore, but it would require non-trivial modifications to the decoding pipeline, and is not guaranteed to work since enforcing strong dissimilarity may break the patterns learned by each head when the diversity coefficient λ is too large.
> > >
> > > > Could the LoRA-MLE baseline be underperforming because the large-rank setup ($rank = K \times r$) leads to optimization instability, while other smaller ranks might perform better?
> > >
> > > In the original submission we provided the performance of the baseline **both** with rank $r$ and $K r$; {8, 40} in audio captioning and {8,24} in image captioning, to account for that. The results for LoRA-MLE with the low rank value $r = 8$ are depicted with diamond markers in Figure 2 and the rows marked with $†$ in Table 2.
> > >
> > >
> > > **References**
> > >
> > > [A] Patel, A. et al. "Styledistance: Stronger content-independent style embeddings with synthetic parallel examples." NAACL-HLT, 2025.
> > >
> > > [B] Zhu, Y. et al. "Diffusion-based diverse audio captioning with retrieval-guided Langevin dynamics." Information Fusion, 2025.
> > >
> > > [C] Beyer, L. et al. "Paligemma: A versatile 3b vlm for transfer." arXiv:2407.07726, 2024.
> > >
> > > [D] Zhang, Y. et al. "Generating Accurate and Diverse Audio Captions Through Variational Autoencoder Framework." IEEE SPL, 2024.
> > >
> > > [E] Zhang, W. et al. "Magic: Multimodal relational graph adversarial inference for diverse and unpaired text-based image captioning." AAAI, 2022.
> > >
> > > [F] Xu, G. et al. "Towards accurate text-based image captioning with content diversity exploration." CVPR, 2021.
> > >
> > > [G] Guzman-Rivera, A. et al. "Efficiently enforcing diversity in multi-output structured prediction."" In AISTATS, pp. 284–292, 2014

---

### Author Response · Authors · 2025-11-22

We would like to thank the reviewers i3JF, xYGJ, i36m, and MhHA [R1, R2, R3, R4] for their valuable remarks and suggestions, which will help us improve the quality of the paper.

Below, we summarize the main changes that will be incorporated in the next revision, in line with the reviewers’ feedback, as well as the results presented in the rebuttal:

* To strengthen the claims of the paper, we will incorporate additional comparisons with state-of-the-art in audio and image captioning [R1, R3, R4], in particular DAC-RLD [A] and PaliGemma-3B [B], as well as the Multi-head approach [R3].

* Further experiments will be included in the paper, such as ablations on the relaxation parameters (ε and the decay rate ρ of the temperature scheduler) [R4], additional values of the rank $r$ [R2], as well as additional metrics, such as Self-CiDEr for diversity and LLM-as-a-judge for quality [R3]. Additionally, quantification the specialization of the hypotheses through linear separability measure in Sentence BERT embedding space will be integrated [R1].

* Finally, we will clarify confusing points [R1, R2, R3, R4], and incorporate a more detailed discussion of potential future work [R1, R3, R4].

[A] Zhu, Y. et al. "Diffusion-based diverse audio captioning with retrieval-guided Langevin dynamics." Information Fusion, 2025.

[B] Beyer, L. et al. "Paligemma: A versatile 3b vlm for transfer." arXiv:2407.07726, 2024.

---

### Meta-Review · Area_Chair_52aB · 2026-01-07

**Summary:**

Reviewers expressed mixed views: strengths in theoretical motivation and efficient handling of ambiguity for diverse generation; concerns about limited novelty (reformulating existing LoRA/MCL ideas), insufficient comparisons to task-specific SOTA (e.g., in captioning/MT), narrow evaluation (few diversity/quality metrics, no text-only LM benchmarks), hyperparameter sensitivity, and mixed empirical gains over baselines.

The presence of a strong reject, combined with the absence of unanimous enthusiasm among accepting reviewers and reliance on planned rather than realized modifications (in paper), leads the AC to decide against acceptance. The AC encourages the authors to incorporate the reviewer feedback and promised enhancements, and to submit the strengthened version to a future conference.

**Reviewer Concerns:**

Mitigated or partially addressed by rebuttal:

- SOTA comparisons: Authors provided results vs. DAC-RLD and PaliGemma-3B, plus multi-head baselines.
- Evaluation breadth: Promised Self-CIDEr, LLM-as-a-judge, specialization quantification (SVM on embeddings), additional ablations (ranks, relaxation params).
- Decoding strategies and rank effects: Clarified beam search usage and provided low-rank baseline results.
- Inference details and specialization evidence: Explained independent generation and provided quantitative specialization metrics.

Remaining:

- Novelty: Core method seen as straightforward combination of existing MCL and LoRA ideas.
- Empirical strength and evaluation scope: Mixed gains persist in some tasks/metrics; broader generality (e.g., text-only LM benchmarks) unaddressed; key improvements rely on promised but unimplemented revisions.
- One reviewer (i36m) gives strong reject on these grounds.

**Reviewer Scores:**

- Reviewer i3JF (original 6): Likely remains 6 (lower confidence).
- Reviewer xYGJ (original 8): Likely remains 8 (lower confidence).
- Reviewer i36m (original 2): Likely remains 2 (higher confidence).
- Reviewer MhHA (original 6): Explicitly stated would raise to 7 post-rebuttal but unwilling to give higher (lower confidence).

---

### Decision · Program_Chairs · 2026-01-26

Reject